# Representation of solutions of second-order linear equations in Barron space via Green's functions

## Abstract

AI-based methods for solving high-dimensional partial differential equations (PDEs) have garnered significant attention as a promising approach to overcoming the curse of dimensionality faced by traditional techniques. This work establishes complexity estimates for the Barron norm of solutions of $d$-dimensional second-order linear PDEs, explicitly capturing the dependence on dimension. By leveraging well-developed theory for elliptic and parabolic equations, we represent the solutions of second-order linear equations using Green's functions. From these representations, we derive complexity bounds for the Barron norm of the solutions. Our results extend the prior work of Chen et al. (2021) in two key aspects. First, we consider more general elliptic and parabolic equations; specifically, we address both time-independent and time-dependent equations. Second, we provide sufficient conditions on the coefficients of the PDEs under which the solutions belong to Barron space rather than approximating the solutions via Barron functions in the $H^1$ norm. As a result, our approach yields theoretically improved results, providing a more intuitive understanding when approximating the solutions of PDEs via two-layer neural networks.

## 1 Introduction and main results

### 1.1 Introduction

Partial differential equations (PDEs) are foundational equations used in modeling complex phenomena across numerous fields, including physics, social sciences, and engineering. Traditionally, numerical methods such as the finite element method (FEM) and the finite difference method (FDM) have been the primary tools for solving PDEs numerically. However, these methods suffer from the curse of dimensionality, which is an exponential increment of computational cost as the dimension of the equation increases. Motivated by these interesting results, we investigate whether a neural network can effectively approximate the solutions of given PDEs without suffering from the curse of dimensionality.

The theoretical analysis of PDEs has traditionally relied on classical function spaces, such as Sobolev spaces ($W^{k,p}(\mathbb{R}^d)$) and Hölder spaces ($C^{k,\alpha}(\mathbb{R}^d)$). However, approximating functions in these spaces with neural networks requires $O(\varepsilon^{-d/k})$ parameters to achieve an accuracy of $\varepsilon$ (Gühring et al., 2020; Lu et al., 2021; Yarotsky, 2017; Yarotsky & Zhevnerchuk, 2020). This indicates that approximating classical function classes by neural networks can be challenging and suffers from the curse of dimensionality. This limitation has motivated the search for alternative function spaces better suited for neural network approximations.

The Barron space, introduced in the seminal work of Barron (1993), is a potential alternative that consists of a collection of all two-layer functions. Approximating functions in Barron space using neural networks can achieve dimension-free convergence rates. These spaces have recently attracted significant attention for their applicability in the approximation theory of two-layer neural networks, also known as shallow networks. Shallow networks play an essential role in developing diverse machine-learning theories. Notable research areas are memory capacities, the representer theorem, and the cost of representation, which are discussed in (Ardeshir et al., 2023; Madden &

Thrampoulidis, 2024; Parhi & Nowak, 2021). In the context of Physics-Informed Neural Networks (PINNs), several advancements have been made in utilizing two-layer network structures to solve PDEs, as shown in (Gao et al., 2023; Li et al., 2023; Xu et al., 2024). Building upon this line of research, estimates of the Barron norm for solutions of PDEs with explicit dependency on the dimension provide significant theoretical value.

Chen et al. (2021) establishes that if the coefficients and the source term of an elliptic PDE lie in Barron space, then the solution of the PDE is $\varepsilon$-close to a Barron function in the Sobolev norm. This result has been extended to nonlinear elliptic equations under linear growth conditions in (Marwah et al., 2023). Additionally, Weinan & Wojtowytsch (2022) demonstrate that several model equations belong to Barron space under appropriate assumptions through representation formulas. They also provide counterexamples where the solutions of PDEs do not belong to the Barron space.

Despite these intriguing results, research on the regularity of time-dependent equations, such as parabolic equations, within the framework of Barron space remains scarce. To the best of our knowledge, Barron norm estimates for the solution of parabolic equations have only been studied in Weinan & Wojtowytsch (2022) for the heat equation. The main reason for the scarcity of results is that the theoretical analysis of parabolic equations presents several key challenges, including time dependencies, handling function spaces involving time, and the anisotropic nature of parabolic equations. We have leveraged and adapted recent advancements in both elliptic and parabolic regularity theory to address these difficulties.

Regularity theory for elliptic and parabolic equations aims to understand the smoothness of solutions under various assumptions regarding the coefficients, forcing terms, and boundary conditions. For a comprehensive overview of the regularity theory of PDEs, we refer the reader to (Gilbarg et al., 1977; Ladyženskaja, 1968). Among the many techniques in the regularity theory of PDEs, the estimates of Green's functions for elliptic and parabolic equations are the primary technique used in this work.

The Green's functions of linear differential operators, denoted by $\mathscr{N}$, are solutions of $\mathscr{N}G = \delta$, where $\delta$ is the Dirac delta function. Section 2.2 provides a detailed definition of these operators for parabolic and elliptic equations. Taking advantage of linearity of the operator, the solution of $\mathscr{N}u = f$ can be represented as $u = f * G$. The construction of Green's functions for elliptic and parabolic equations has been extensively studied. See for instance (Grüter & Widman, 1982; Littman et al., 1963; Kim & Sakellaris, 2019; Hofmann & Lewis, 2001; Davey et al., 2018) for elliptic equations, and (Nash, 1958; Moser, 1964; Cho et al., 2008; Semenov, 2006; Qian & Xi, 2019; Kim & Xu, 2021) for parabolic equations under various assumptions such as coefficients, domain regularity and so on. In this study, we primarily build upon the methodologies proposed by (Kim & Xu, 2021).

The main contributions of this paper are summarized as follows.

1. **Barron norm complexity estimates for time-dependent parabolic equations:** We establish, for the first time, estimates on the complexity of the Barron norm for the solutions of general second-order linear parabolic equations the explicit dependency on the dimension.

2. **Improvement of theoretical results for elliptic equations:** Even for time-independent elliptic equations, our results represent a significant advancement over existing work (Chen et al., 2021). We establish that, under appropriate assumptions, the solutions of these PDEs belong directly to Barron space rather than merely approximating them in the Sobolev sense within Barron space. Please refer to Remark 3. This refinement constitutes a theoretical improvement, offering deeper insights into the structure of the solutions.

3. **Theoretical foundation:** Our research provides a theoretical foundation demonstrating that AI-based PDE solvers or Green's function learning methods can represent solutions without suffering from the curse of dimensionality.

Before introducing the results of this paper, we remind the reader that all mathematical notations are summarized in Appendix A.

## 1.2 BARRON NORM ESTIMATES FOR PARABOLIC EQUATIONS

Our first main result concerns parabolic equations. To this end, for $\mathbf{A}(t,x) \in \mathbb{R}^{d \times d}, \mathbf{b}(t,x) \in \mathbb{R}^d$ and $\mathbf{c}(t,x) \in \mathbb{R}$, we define a parabolic operator, $\mathcal{P}$ as follows:

$$
\begin{aligned}
\mathcal{P}u :=& \partial_t u - \sum_{i,j=1}^{d} D_i \left( a^{ij}(t,x) D_j u + b_i(t,x) u \right) + \sum_{i=1}^{d} c_i(t,x) D_i u + d(t,x) u, \\
=& \partial_t u - \operatorname{div}\left( \mathbf{A}(t,x) Du \right) + \mathbf{b}u \right) + \mathbf{c} \cdot Du + d(t,x) u.
\end{aligned}
\tag{1}
$$

Throughout this work, we denote $\mathscr{D} = (0,t) \times \mathbb{R}^d \subset \mathbb{R}^{d+1}$, which is a cylindrical domain with $0 < t < \infty$. We assume that the coefficients satisfy the following conditions.

**Assumptions 1.** *Suppose that the coefficients on $\mathcal{P}$ satisfy the following assumptions:*

*(1) The matrix $[\mathbf{A}(t,x)]^{ij} = (a^{ij}(t,x))$ is a symmetric matrix, i.e., $a^{ij}(t,x) = a^{ji}(t,x)$, satisfying the following ellipticity and boundedness conditions for some $0 < \lambda \leq \Lambda < \infty$:*

$$
\lambda |\xi|^2 \leq \sum_{i,j=1}^{d} a^{ij}(t,x) \xi_i \xi_j \quad \text{and} \quad \sum_{i,j=1}^{d} |a^{ij}(t,x)|^2 \leq \Lambda^2.
$$

*(2) We assume that the coefficients $\mathbf{A}(t,x), \mathbf{b}(t,x), \mathbf{c}(t,x)$ and $d(t,x)$ belong to the VMO($\mathbb{R}^{d+1}$) space (Defintion 4). That is, we assume that*

$$
\lim_{r \to 0} \sup_{D_r \subset \mathbb{R}^{d+1}} \frac{1}{|D_r|} \int_{D_r} |\mathbf{A}(t,x) - (\mathbf{A})_{D_r}|^2 \, dx \, dt = 0,
$$

$$
\lim_{r \to 0} \sup_{D_r \subset \mathbb{R}^{d+1}} \frac{1}{|D_r|} \int_{D_r} |\mathbf{b}(t,x) - (\mathbf{b})_{D_r}|^2 \, dx \, dt = 0,
$$

$$
\lim_{r \to 0} \sup_{D_r \subset \mathbb{R}^{d+1}} \frac{1}{|D_r|} \int_{D_r} |\mathbf{c}(t,x) - (\mathbf{c})_{D_r}|^2 \, dx \, dt = 0,
$$

$$
\lim_{r \to 0} \sup_{D_r \subset \mathbb{R}^{d+1}} \frac{1}{|D_r|} \int_{D_r} |d(t,x) - (d)_{D_r}|^2 \, dx \, dt = 0.
$$

*In this work, $D_r(t_0, x_0) = (t_0 - r^2, t_0 + r^2) \times C_r(x_0)$, where $C_r(x_0)$ denotes a cube with side length $2r$ and center at $x_0$. When it is clear from the context or not important, we shall omit the points $t_0$ and $x_0$.*

*(3) We assume that in a distribution sense*

$$
d(t,x) - \operatorname{div} \mathbf{b}(t,x) \geq 0
$$

*and*

$$
\int_{\mathscr{D}} (c^i(t,x) - b^i(t,x))) \varphi_1 D_i \varphi_2 \, dx \, dt \geq 0
$$

*for all smooth functions satisfying $\varphi_1, \varphi_2 \geq 0$.*

*(4) We assume that*

$$
\mathbf{b}(t,x), \mathbf{c}(t,x) \in L_\infty(\mathscr{D}) \cap L_1(\mathscr{D}) \quad \text{and} \quad d(t,x) \geq 0, d(t,x) \in L_\infty(\mathscr{D}).
$$

For a parabolic operator $\mathcal{P}$, as defined in equation 1, let $u \in \mathring{V}_2(\mathscr{D})$ be a weak solution of

$$
\begin{cases}
\mathcal{P}u = f & \text{in } \mathscr{D}, \\
u(0, \cdot) = g(\cdot) & \text{on } \mathbb{R}^d.
\end{cases}
\tag{2}
$$

The definition of a weak solution is stated below.

**Definition 1** (Weak solution of equation 2)**.** *Let $\mathscr{D} = (0, t) \times \mathbb{R}^d$ and $f \in L_{\overline{p}', \overline{q}'}(\mathscr{D})$, where $\overline{p}'$ and $\overline{q}'$ are Hölder conjugates of $\overline{p}$ and $\overline{q}$, respectively, and $\overline{p}$ and $\overline{q}$ satisfy $d/\overline{p} + 2\overline{q} = d/2$ with ranges specified in equation 18. We say that $u \in \mathring{V}_2(\mathscr{D})$ is a weak solution of equation 2 if for almost all $t_1 \in (0, t)$ the identity*

$$
I(t_1; u, \phi) := \int_{\mathbb{R}^d} u(t_1, x)\phi(t_1, x)\, dx - \int_0^{t_1} \int_{\mathbb{R}^d} u\partial_t \phi\, dx\, dt
$$
$$
+ \int_0^{t_1} \int_{\mathbb{R}^d} (a^{ij}D_j u + b^i u)D_i\phi\, dx\, dt + \int_0^{t_1} \int_{\mathbb{R}^d} (c^i D_i u + du)\phi\, dx\, dt \qquad (3)
$$
$$
- \int_0^{t_1} \int_{\mathbb{R}^d} f\phi\, dx\, dt = \int_{\mathbb{R}^d} g(x)\phi(0, x)\, dx
$$

*holds for all $\phi \in C_c^{1,1}(\mathscr{D})$.*

**Remark 1.** *It is worth mentioning that if we choose a test function in $C_c^{1,1}(\overline{\mathscr{D}} \setminus \partial_p \mathscr{D})$, then equation 3 changes to:*
$$
I(t_1; u, \phi) = 0, \quad \forall \phi \in C_c^{1,1}(\mathscr{D} \setminus \partial_p \mathscr{D}).
$$

**Theorem 1.** *Suppose that $\mathcal{P}$ is a parabolic operator defined in equation 1 with Assumption 1, $f \in L_{\frac{2d+2}{d+4}}(\mathscr{D}) \cap \mathcal{B}(\mathscr{D})$ and $g \in L_2(\mathbb{R}^d) \cap \mathcal{B}(\mathbb{R}^d)$. If the weak solution $u \in \mathring{V}_2(\mathscr{D})$ is the continuous function, then we have*

$$
\|u(t, \cdot)\|_{\mathcal{B}(\mathbb{R}^d)}
$$
$$
\leq 2(1 + \|c - b\|_{L^\infty(\mathbb{R}^d)} + \Lambda^{\frac{1}{2}}\sqrt{t})\|g\|_{\mathcal{B}(\mathbb{R}^d)}
$$
$$
+ 2\|f\|_{B((0,t)\times\mathbb{R}^d)}\left(t + \frac{t^2}{2} + \Lambda^{\frac{1}{2}}\left(\frac{2t^{\frac{3}{2}}}{3} + \frac{2t^{\frac{5}{2}}}{5}\right) + \|c - b\|_{L^\infty(\mathscr{D})}\left(\frac{t^2}{2} + \frac{t^3}{6}\right)\right).
$$

The theorem states that if $0 < \Lambda < \infty$ and $u$ is a continuous function, then the curse of dimensionality can be avoided when approximating $u(\cdot, t)$ using two-layer networks. Additionally, it asserts that the Barron norm of $u(\cdot, t)$ grows at most at a rate of $t^3$. For the comparison with the model equations, we refer to Appendix N.

## 1.3 BARRON NORM ESTIMATES FOR ELLIPTIC EQUATIONS

The second main result is to extend the existing results for elliptic equations. Let $\mathcal{L}$ be an elliptic operator defined as

$$
\mathcal{L}u := -\sum_{i,j=1}^d D_i\left(a^{ij}(x)D_j u\right) + d(x)u, \qquad (4)
$$
$$
= -\operatorname{div}\left(\mathbf{A}(x)Du\right) + d(x)u.
$$

Note that we choose $\mathbf{b} = \mathbf{c} = 0$ and remove $\partial_t u$ in $\mathcal{P}$ to get $\mathcal{L}$.

**Assumptions 2.** *Suppose that the coefficients on $\mathcal{L}$ satisfy the following assumptions:*

*(1) The matrix $[\mathbf{A}(x)]^{ij} = (a^{ij}(x))$ is a symmetric matrix, i.e., $a^{ij}(x) = a^{ji}(x)$, satisfying the following ellipticity and boundedness condition for some $0 < \lambda \leq \Lambda < \infty$:*

$$
\lambda|\xi|^2 \leq \sum_{i,j=1}^d a^{ij}(x)\xi_i\xi_j \quad \text{and} \quad \sum_{i,j=1}^d |a^{ij}(x)|^2 \leq \Lambda.
$$

*(2) We assume that the coefficients $\mathbf{A}(x)$ and $d(x)$ belongs to the $\mathrm{VMO}(\mathbb{R}^d)$ space :*

$$
\lim_{r \to 0} \sup_{C_r \subset \mathbb{R}^d} \frac{1}{|C_r|} \int_{C_r} |\mathbf{A}(x) - (\mathbf{A})_{C_r}|^2\, dx = 0,
$$

$$
\lim_{r \to 0} \sup_{C_r \subset \mathbb{R}^d} \frac{1}{|C_r|} \int_{C_r} |d(x) - (d)_{C_r}|^2\, dx = 0.
$$

*(3) There exists $d_{min}, d_{max} > 0$ satisfying $d_{min} \le d(x) \le d_{max}$.*

Below, we introduce the equations considered in this subsection.

$$\mathcal{L}u = f \quad \text{in} \quad \mathbb{R}^d. \tag{5}$$

The weak solution of the elliptic equation is defined similarly to the parabolic equations.

**Definition 2** (Weak solution of equation 5). *Suppose that $f \in L_{2^*}(\mathbb{R}^d)$, where $2^* = 2d/(d-2)$. We say that $u \in \mathring{W}_2^1(\mathbb{R}^d)$ is a weak solution of equation 5 if the identity*

$$J(u, \phi) := \int_{\mathbb{R}^d} a^{ij} D_j u D_i \phi \, dx + \int_{\mathbb{R}^d} du \phi \, dx \, dt - \int_{\mathbb{R}^d} f \phi \, dx \, dt = 0$$

*holds for all $\phi \in C_c^{1,1}(\mathscr{D})$.*

The following theorem is the second main theorem of this work.

**Theorem 2** (Barron norm estaimtes). *Suppose that $\mathcal{L}$ is an elliptic operator defined in equation 4, with Assumption 2. We further assume that $f \in L_{\frac{2d}{d+2}}(\mathbb{R}^d) \cap \mathcal{B}(\mathbb{R}^d)$. Then if the weak solution $u \in W_2^1(\mathbb{R}^d)$ is a continuous function, then $u$ belongs to the Barron space with the estimate*

$$\|u\|_{\mathcal{B}(\mathbb{R}^d)} \le 2 \left( \frac{1}{d_{min}} + \frac{\Lambda^{\frac{1}{2}} \sqrt{\pi} \Gamma(\frac{d+1}{2})}{2 d_{min}^{\frac{3}{2}} \Gamma(\frac{d}{2})} \right) \|f\|_{\mathcal{B}(\mathbb{R}^d)}.$$

**Remark 2.** *Before we proceed further, let us highlight add some remarks on the assumptions Assumption 1, Assumption 2 and notations.*

- *In the context of the regularity theory of elliptic and parabolic equations, VMO or small BMO coefficients are extensively studied topics. For instance, we refer (Byun & Wang, 2004; Dong & Kim, 2010; Bramanti & Cristina Cerutti, 1993) for details. VMO space includes a wide range of functions, including uniformly continuous functions. Also, there exists a function in VMO that has a discontinuity. Therefore in the local sense, Barron function belgons to thenVMO space. For the overall remark on the VMO space, we refer to Appendix D.*

- *The Assumption 1 - (4) and Assumption 2-(4) is to ensure that $I(t_1, u, u) < \infty$ with $u \in \mathring{V}_2(\mathscr{D})$ in the parabolic equations and $J(u, u) < \infty$ with $u \in W_2^1(\mathbb{R}^d)$ in the elliptic equations. Different assumptions may be imposed on the coefficients of the lower-order terms, but the exponents in Assumption 1 -(2) and Assumption 2 - (2) need to be changed.*

- *By the Assumption 1 - (3), we have the energy estimates, Lemma 6 where the constant $C_{energy}$ does not depend on the $\mathbf{b}, \mathbf{c}$ and $d(t, x)$, which is crucial crucial for establishing the existence and the uniqueness of the weak solution.*

- *The letter $d$ is used for the dimension and the coefficient functions for $\mathcal{P}$ and $\mathcal{L}$. To avoid confusion, we shall use notations such as $d(t, x), d(x), d_\delta(t, x), d_\delta(x)$ if it is not a constant, and $\overline{d}$ if it is a constant coefficient.*

**Remark 3** (Comparison with the result of Chen et al. (2021)). *The main difference from the result in (Chen et al., 2021) is worth mentioning. The Barron space used in (Chen et al., 2021) is slightly different from ours, so we shall refer to it as $\tilde{\mathcal{B}}$. Chen et al. (2021) assume the followings:*

- *Regularity assumptions: $\mathbf{A}(x), d(x), f(x) \in \tilde{\mathcal{B}}$,*

- *An activation function $\sigma$ is chosen such that the function $h(y_1, y_2) = \sigma(y_1)\sigma(y_2)$ belongs to the Barron space in $\mathbb{R}^2$,*

- *$\sigma'(\cdot)$ and $\sigma''(\cdot)$ belong to the Barron space, $\tilde{\mathcal{B}}$.*

*Then, for all $\varepsilon \in (0, 1/2)$, there exists $u_m \in \tilde{\mathcal{B}}$ satisfying*

$$\|u_m - u\|_{W_2^1(\mathbb{R}^d)} \le \varepsilon \quad \text{and} \quad \|u_m\|_{\mathcal{B}} \le C_1 \left( \frac{d}{\varepsilon} \right)^{C_2 |\log \varepsilon|}.$$

*In this work, we assume that the coefficients are not required to belong to $\mathcal{B}(\mathbb{R}^d)$. We assume the activation function is $\sigma(x) = ReLU(x)$, which does not satisfy the multiplicity condition. We then demonstrate that $u \in \mathcal{B}(\mathbb{R}^d)$, without encountering the curse of dimensionality in the estimates, provided that $u$ is a continuous solution. For remarks on the continuity of the solution, we refer to the following Remark and Appendix E. For technical remarks on the activation functions, we refer to Appendix F.*

**Remark 4** (Continuity of the weak soltuion)**.** *The continuity of the solution holds in many cases, even with minimal assumptions on the regularity of elliptic and parabolic equations. For this reason, we believe our approach significantly advances the theory of existing research and can also be effectively applied to parabolic equations. We summarize some well-known conditions under which weak solutions of elliptic and parabolic equations are continuous in Appendix E.*

## 2 PRELIMINARIES

In this section, we introduce the preliminaries used in the subsequent sections.

### 2.1 BARRON SPACE

Roughly speaking, the Barron space is a function space comprising all two-layer networks, first introduced by Barron (1993). There are many variations, and precise definitions depend on the context. For the exploration of the mathematical properties of Barron space and its generalizations, we refer to (Wojtowytsch et al., 2021; Chen, 2024; Ma et al., 2022; Meng & Ming, 2022; Xu, 2020; Wu, 2023). We shall focus on the ReLU activation function for technical reasons. For remarks on the Barron space with differnt activation functions, we refer to Appendix F.

For a choosen activation $\sigma : \mathbb{R} \to \mathbb{R}$ and a probability distribution $\pi$ over the parameters $(a, w, b) \in \mathbb{R} \times \mathbb{R}^d \times \mathbb{R}$, we denote

$$f_\pi(x) = \int_{\mathbb{R}^{d+2}} a\,\sigma(w^\top x + b)\,\pi(da \otimes dw \otimes db).$$

For any subset $\Omega \subset \mathbb{R}^d$ and $f : \Omega \to \mathbb{R}$, we define a set of probability distributions as

$$A_f(\Omega) := \{\pi \in \mathbb{P}(\mathbb{R} \times \mathbb{R}^d \times \mathbb{R}) : f_\pi|_\Omega = f\}.$$

Here, $\mathbb{P}(X)$ denotes a probability space on $X$. Then we define the Barron space $\mathcal{B}(\Omega)$ equipped with the norm $\|f\|_{\mathcal{B}(\Omega)}$ as

$$\|f\|_{\mathcal{B}(\Omega)} := \inf \left\{ \int_{\mathbb{R}^{d+2}} |a|(\|w\| + |b|)\,\pi(da \otimes dw \otimes db) : \pi \in A_f(\Omega) \right\}. \tag{6}$$

We say that $f \in \mathcal{B}(\Omega)$ if $\|f\|_{\mathcal{B}(\Omega)}$ is finite. The main property of the Barron space is that if $f \in \mathcal{B}(\Omega)$, then it can be approximated by the two-layer network with the rate of $O(m^{-1/2})$, independent of the dimension $d$. We provide the statement of the theorems from (Ma et al., 2022, Section 2.2)

**Proposition 1.** *For any $f \in \mathcal{B}(\Omega)$ and $m > 0$, there exists a two-layer neural network*

$$f_m(x; \Theta) = \frac{1}{m} \sum_{k=1}^{m} a_k \sigma(w_k^\top x + b_k),$$

*(Here, $\Theta$ denotes the parameters $\{(a_k, \mathbf{b}_k, c_k), k \in 1, \cdots, m\}$ in the neural network), such that*

$$\|f(\cdot) - f_m(\cdot; \Theta)\|_{L_2(\Omega)}^2 \leq \frac{3\|f\|_{\mathcal{B}(\Omega)}^2}{m}.$$

*Furthermore, we have*

$$\|f_m(\cdot; \Theta)\|_{path} := \frac{1}{m} \sum_{j=1}^{m} |a_j|(\|w_j\|_1 + |b_j|) \leq 2\|f\|_{\mathcal{B}(\Omega)}. \tag{7}$$

*Here, $\|\cdot\|_{path}$ is called a path norm.*

**Proposition 2.** *Let us denote*

$$\mathcal{N}_Q = \left\{ \frac{1}{m} \sum_{k=1}^m a_k \sigma(w_k^\top x + b_k) : \frac{1}{m} \sum_{k=1}^m |a_k|(\|w_k\|_1 + |b_k|) \le Q, m \in \mathbb{N}^+ \right\}.$$

*Let $f^* : \Omega \to \mathbb{R}$ be a continuous function. Assume that there exists a constant $Q > 0$ and a sequence of functions $(f_m) \subset \mathcal{N}_Q$ such that*

$$f_m(\mathbf{x}) \to f^*(\mathbf{x}), \quad \forall \mathbf{x} \in \Omega.$$

*Then there exists a probability distribution $\rho^* \in \mathbb{P}(\mathbb{R} \times \mathbb{R}^d \times \mathbb{R})$, such that*

$$f^*(\mathbf{x}) = \int a\sigma(w^\top x + b)\, \rho^*(da, dw, db), \quad \forall \mathbf{x} \in \Omega.$$

*Furthermore, we have $f^* \in \mathcal{B}(\Omega)$ with $\|f^*\|_{\mathcal{B}(\Omega)} \le Q$.*

## 2.2 GREEN'S FUNCTIONS

Green's functions provide a powerful method for solving linear differential equations. For the linear parabolic operator $\mathcal{P}$ defined in equation 1, the Green's function is defined as $G(X, Y) = G(t, x, s, y) : \mathscr{D} \times \mathscr{D} \to \mathbb{R}$ satisfying:

$$\begin{cases} \mathcal{P}G(\cdot, \cdot, s, y) = 0 & \text{in} \quad (s, \infty) \times \mathbb{R}^d, \\ G(t, x, s, y) = \delta(x - y) & \text{on} \quad \{t = s\} \times \mathbb{R}^d, \end{cases}$$

where $\delta$ is the Dirac delta function. The solution of equation 2 can be represented as

$$u(t, x) = \int_{\mathbb{R}^d} G(t, x, 0, y)g(y)\, dy + \int_0^t \int_{\mathbb{R}^d} G(t, x, s, y)f(s, y)\, dy\, ds. \tag{8}$$

For the linear elliptic operator $\mathcal{L}$, defined in equation 4, the Green's function $G(x, y) : \mathbb{R}^d \times \mathbb{R}^d \to \mathbb{R}$ satisfies

$$\mathcal{L}G(x, y) = \delta(x - y).$$

The solution of equation 5 can then be expressed as:

$$u(x) = \int_{\mathbb{R}^d} G(x, y)f(y)\, dy.$$

# 3 PROOF OF THE MAIN THEOREM

This section briefly sketches the proofs of the theorems in Section 1.3 and Section 1.2. Detailed proofs are provided in the Appendix.

## 3.1 APPROXIIMATING THE PARABOLIC EQUATIONS AND RELATED GREEN'S FUNCTIONS

Let us denote

$$\mathcal{P}_{const} := \partial_t u - \overline{a}^{ij} D_{ij}^2 u + (\overline{c}^i - \overline{b}^i)D_i u + \overline{d}u,$$

where $\overline{A}^{ij} = \overline{a}^{ij}$ is the constant positive definite symmetric matrix and $\overline{\mathbf{b}} = (\overline{b}^i, \cdots, \overline{b}^d), \overline{\mathbf{c}} = (\overline{c}^i, \cdots, \overline{c}^d)$ are the constant vectors and $\overline{d}$ is the constant coefficients. The following lemma provides an explicit formula for the Green's function for $\mathcal{P}_{const}$.

**Lemma 1.** *The Green's function $G(t, x, s, y)$ of the following equation*

$$\begin{cases} \mathcal{P}_{const}u = 0 & \text{in} \quad (s, \infty) \times \mathbb{R}^d \\ u(t, x) = \delta(x - y) & \text{on} \quad \{t = s\} \times \mathbb{R}^d. \end{cases}$$

*is*

$$G_{const}(t, x, s, y) = \Phi_{const}(t - s, x - y),$$

*where*

$$\Phi_{const}(t, x) = \frac{1}{(4\pi t)^{d/2}|\det(\overline{A})|^{1/2}} \exp\left\{ -\frac{\langle \overline{A}^{-1}(x + (\overline{c} - \overline{b})t), x + (\overline{c} - \overline{b})t \rangle}{4t} - \overline{d}t \right\}.$$

A direct calculation can derive the proof. We then define a parabolic operator $\mathcal{P}_\delta$ that approximates $\mathcal{P}$. To this end, for each $\delta > 0$, there exists $R_\delta$ satisfying

$$\frac{1}{|D_{R_\delta}|} \int_{D_{R_\delta}} |A(t,x) - (A)_{D_{R_\delta}}|^2 \, dx \, dt \le \delta, \qquad \frac{1}{|D_{R_\delta}|} \int_{D_{R_\delta}} |\mathbf{b}(t,x) - (\mathbf{b})_{D_{R_\delta}}|^2 \, dx \, dt \le \delta,$$

$$\frac{1}{|D_{R_\delta}|} \int_{D_{R_\delta}} |\mathbf{c}(t,x) - (\mathbf{c})_{D_{R_\delta}}|^2 \, dx \, dt \le \delta, \qquad \frac{1}{|D_{R_\delta}|} \int_{D_{R_\delta}} |d(t,x) - (d)_{D_{R_\delta}}|^2 \, dx \, dt \le \delta,$$

for any $D_{R_\delta} \subset \mathbb{R}^{d+1}$. Then, we divide $\mathscr{D}$ by the cubes $D_{R_\delta}$ without overlapping and excluding measure zero sets. We denote

$$\mathscr{D} = \bigcup_k D_{R_\delta, k} \cup \mathcal{N},$$

where $\mathcal{N}$ is a measure zero set. Then, we define averaged coefficients as

$$a_\delta^{ij}(t,x) := \sum_k (a^{ij}(t,x))_{D_{R_\delta}^k} \chi_{D_{R_\delta}^k}, \qquad b_\delta^i(t,x) := \sum_k (b^i(t,x))_{D_{R_\delta}^k} \chi_{D_{R_\delta}^k},$$

$$c_\delta^i(t,x) := \sum_k (c^i(t,x))_{D_{R_\delta}^k} \chi_{D_{R_\delta}^k}, \qquad d_\delta^{ij}(t,x) := \sum_k (d(t,x))_{D_{R_\delta}^k} \chi_{D_{R_\delta}^k}.$$

If $(t,x) \in \mathcal{N}$, we define $a_\delta^{ij}(t,x), b_\delta^i(t,x), c_\delta^i(t,x)$ and $d_\delta(t,x)$ arbitrary. We then define the "approximated" parabolic equation as follows:

$$\begin{aligned} \mathcal{P}_\delta u :=& \partial_t u - \sum_{i,j=1}^d D_i \left( a_\delta^{ij}(t,x) D_j u + b_\delta^i(t,x) u \right) + \sum_{i=1}^d c_\delta^i(t,x) D_i u + d_\delta(t,x) u, \\ =& \partial_t u - \text{div} \left( \mathbf{A}_\delta(t,x) Du \right) + \mathbf{b}_\delta(t,x) u_\delta) + \mathbf{c}_\delta(t,x) \cdot Du + d_\delta(t,x) u. \end{aligned} \tag{9}$$

We denote $G_\delta(t,x,s,y)$ for the function replacing $\overline{\mathbf{A}}$, $\overline{\mathbf{b}}$, $\overline{\mathbf{c}}$ and $\overline{d}$ by $\overline{\mathbf{A}}_\delta$, $\overline{\mathbf{b}}_\delta$, $\overline{\mathbf{c}}_\delta$ and $\overline{d}_\delta$ in $G_{const}(t,x,s,y)$. We then claim that $G_\delta$ is a fundamental solution function for the operator equation 9. The precise statement is given below.

**Lemma 2.** *Suppose that $f \in C^{2,1}(\mathscr{D}) \cap L_{\frac{2d+4}{d+4}}(\mathscr{D})$ and $g \in C^2(\mathbb{R}^d) \cap L_2(\mathbb{R}^d)$, then*

$$\begin{aligned} u_\delta(t,x) :=& \int_{\mathbb{R}^d} G_\delta(t,x,0,y) * g(y) \, dy + \int_0^t \int_{\mathbb{R}^d} G_\delta(t,x,s,y) * f(s,y) \, dy \, ds \\ =:& u_{\delta,hom}(t,x) + u_{\delta,inhom}(t,x) \end{aligned} \tag{10}$$

*is the weak solution of*

$$\begin{cases} \mathcal{P}_\delta u = f & in \quad \mathscr{D}, \\ u(0,x) = g(x) & on \quad \mathbb{R}^d. \end{cases} \tag{11}$$

The proof of Lemma 2 is provided in Appendix J.

### 3.2 BARRON NORM ESTIMATE OF APPROXIMATED EQUATIONS

In this section, we provide proof of the Barron norm estimates for the function defined in equation 10.

**Theorem 3.** *Suppose that $u_\delta(t,x)$ is the function defined in equation 10 for some $f \in \mathcal{B}((0,\infty) \times \mathbb{R}^d)$ and $g \in \mathcal{B}(\mathbb{R}^d)$. Then, we have*

$$\begin{aligned} &\|u_\delta(t,\cdot)\|_{\mathcal{B}(\mathbb{R}^d)} \\ &\le 4(1 + \|c - b\|_{L^\infty(\mathbb{R}^d)} + \Lambda^{\frac{1}{2}} \sqrt{t}) \|g\|_{\mathcal{B}(\mathbb{R}^d)} \\ &+ 2\|f\|_{B((0,t)\times\mathbb{R}^d)} \left( t + \frac{t^2}{2} + \Lambda^{\frac{1}{2}} \left( \frac{2t^{\frac{3}{2}}}{3} + \frac{2t^{\frac{5}{2}}}{5} \right) + \|c - b\|_{L^\infty(\mathscr{D})} \left( \frac{t^2}{2} + \frac{t^3}{6} \right) \right). \end{aligned} \tag{12}$$

The proof can be found in Section K.

## 3.3 COMPARISON ESTIMATES

Next, we show that the weak solution of equation 11 and the weak solution of equation 2 are close in a local sense.

**Theorem 4.** *Let $u$ be the weak solution of equation 2 and $u_\delta$ be the weak solution equation 11. Then, for all $\delta > 0$, there exists some $\alpha = \alpha(d, \lambda, \Lambda)$ and constant $\tilde{C} > 0$ satisfying*

$$\|u - u_\delta\|_{V_2(B_R)} \le \frac{\tilde{C}_1}{R} + \tilde{C}_2 \delta^\alpha.$$

*Here, the constnat $\tilde{C}_1$ depends on $t, d, \lambda, \Lambda, \|f\|_{L_{\frac{2d+4}{d+4}}(\mathscr{D})}, \|g\|_{L_2(\mathbb{R}^d)}$ and constant $\tilde{C}_2$ depends on $R, t, d, \lambda, \Lambda, \|f\|_{L_{\frac{2d+4}{d+4}}(\mathscr{D})}, \|g\|_{L_2(\mathbb{R}^d)}, \|\boldsymbol{b}\|_{L^\infty(\mathscr{D})}, \|\boldsymbol{c}\|_{L^\infty(\mathscr{D})}$ and $\|d\|_{L^\infty(\mathscr{D})}$.*

The proof can be found in Section L

## 3.4 PROOF OF THEOREM 1

*Proof.* We first prove the theorem under the assumption that $f \in C_{2,1}(\mathscr{D}) \cap L_{\frac{2d+4}{d+4}}(\mathscr{D}) \cap \mathcal{B}(\mathscr{D})$ and $g \in C^2(\mathbb{R}^d) \cap L_2(\mathbb{R}^d) \cap \mathcal{B}(\mathbb{R}^d)$. By Lemma 2, $u_\delta(x, t)$ defined in equation 2 is the weak solution of equation 11.

Let us denote $Q_t$ as the upper bound of the Barron norm of $u_\delta(t, \cdot)$, given by the right-hand side of equation 12. For each $k \in \mathbb{N}$ we construct an increasing sequece $\{R_k\}_k$ such that $1/R_k \le 1/(4k\tilde{C}_1)$. By Theorem 4, we choose $\delta_k$ satisfying

$$\|u - u_{\delta_k}\|_{V_2(B_{R_k})} \le \frac{1}{2k}.$$

By Proposition 1, there exists a sequence of two-layer networks $\{u_{\delta_k, m}(t_1, \cdot)\}$ such that

$$\|u_{\delta_k, m}(t_1, \cdot) - u_{\delta_k}(t_1, \cdot)\|_{L^2(B_{R_k})}^2 \le \frac{3Q_t}{m},$$

for some $t_1 \in (0, t)$. Then, choose $m_k \in \mathbb{N}$ such that

$$\|u_{\delta_k, m_k}(t_1, \cdot) - u_{\delta_k}(t_1, \cdot)\|_{L^2(B_{R_k})}^2 \le \frac{1}{2k}.$$

Note that the two-layer networks $\{u_{\delta_k, m_k}(t_1, \cdot)\}_k$ converge to $u(t_1, \cdot)$ in $L^2(B_R)$ for all $R > 0$. Since the path norms of $\{u_{\delta_k, m_k}(t_1, )\}_k$ are bounded by $2Q_t$, the Lipschitz semi-norm of $\{u_{\delta_k, m_k}\}$ are bounded by $2Q_t$. Therefore, $u_{\delta_k, m_k}(t_1, \cdot)$ converges to $u(t_1, \cdot)$ for all $x \in \mathbb{R}^d$. By Proposition 2, $u(t_1, \cdot) \in \mathcal{B}(B_R)$ with the estimate in equation 12.

Next, suppose that $f \notin C^{2,1}(\mathscr{D})$ and $g \notin C^2(\mathbb{R}^d)$. Since $f$ and $g$ still belong to Lebesgue space and Barron space, we use a standard mollifier, defined in (Evans, 2022, Appendix C.5), $\eta_\varepsilon$ to approximate $f$ and $g$ via smooth functions.

We follow the same procedure for $f_\varepsilon = f * \eta_\varepsilon$ and $g_\varepsilon = g * \eta_\varepsilon$. Let $u_\varepsilon$ be the weak solution of equation 2. Then, equation 12 holds with $u, f, g$ replaced by $u_\varepsilon, f_\varepsilon$ and $g_\varepsilon$, respectively. By the properties of the mollifier, $f_\varepsilon \to f$ in $L_{\frac{2d+2}{d+4}}(\mathscr{D})$ and $g_\varepsilon \to g$ in $L_2(\mathbb{R}^d)$. By the energy estimates equation 20, $u_\varepsilon \to u$ in $V_2(\mathscr{D})$ as well. Finally, by the properties of the mollifier, we have

$$\int_{\mathbb{R}^d} \eta_\varepsilon(x)\, dx = 1 \quad \text{and} \quad \int_{\mathbb{R}^d} |x| \eta_\varepsilon(x)\, dx \le C(d)\varepsilon.$$

Using these properties, we have $f_\varepsilon \to f$ in $\mathcal{B}(\mathscr{D})$ and $g_\varepsilon \to g$ in $\mathcal{B}(\mathbb{R}^d)$. Taking $\varepsilon \to 0$ the estimate equation 12 holds without the assumption that $f \in C^{2,1}(\mathscr{D})$ and $g \in C^2(\mathbb{R}^d)$. □

## 3.5 SKETCH OF THE PROOF OF THEOREM 2

Since proving Theorem 2 proceeds similarly, we shall only highlight the difference for simplicity. We define a linear elliptic equation as:

$$\mathcal{L}_\delta u_\delta := -\text{div}\left(\mathbf{A}_\delta(x)Du_\delta\right) + d_\delta u_\delta = f \quad \text{in} \quad \mathbb{R}^d. \tag{13}$$

Then we denote $G_{\delta,zc}(t,x,s,y)$ for the Green's function of the parabolic operator equation 9 with $\mathbf{b} = \mathbf{c} = 0$. Then, with some scaling scalar $C_{scale} > 0$, we have

$$
\begin{aligned}
K_\delta(x,y) &= C_{scale} \int_0^\infty G_{\delta,zc}(t,x,0,y)\,dt \\
&= C_{scale} \int_0^\infty \frac{1}{(4\pi t)^{d/2}|\mathbf{A}_\delta|^{1/2}} \exp\left\{ -\frac{\langle \mathbf{A}_\delta^{-1}(x-y), x-y \rangle}{4t} - d_\delta t \right\} dt
\end{aligned}
\tag{14}
$$

which is a fundamental solution of equation 13. Under appropriate regularity assumptions on $f$, $u_\delta = K_\delta * f$ is the weak solution of equation 13. The proof proceeds similarly to the proof of Lemma 2, which we omit here. Then, the weak solutions of equation 5 and equation 13 are sufficiently close, satisfying

$$
\|u - u_\delta\|_{W_2^1(B_R)} \leq \frac{\tilde{C}_1}{R} + \tilde{C}_2 \delta^\alpha
$$

for some $\alpha > 0$, where $\tilde{C}_1$ does not depend on $R$ and $\tilde{C}_2$ depends on $R$ and $d, \lambda, \Lambda, d_{min}, d_{max}$ and $\|f\|_{L_{2d/(d+2)}(\mathbb{R}^d)}$. Additionally, higher integrability results for elliptic equations can be found, for instance, in (Giusti, 2003, Remark 6.12). This proof is similar to that of Theorem 4; therefore, we omit it in this paper.

The Barron norm estimate of $u_\delta$ is provided in Appendix M. We then complete the proof using the approximation argument outlined in the proof of Theorem 1 substituting Lemma 7, instead of Lemma 6 for the energy estimates in the elliptic equations.

## 4 CONCLUSION AND FUTURE WORKS

In this paper, we have demonstrated that solutions of second-order linear PDEs can be effectively represented in the Barron space using Green's functions. This result theoretically supports existing two-layer PINN methods (Gao et al., 2023; Li et al., 2023; Xu et al., 2024), particularly for high-dimensional problems. Below, we discuss interesting directions for future research.

- Barron norm estimates for the nonlinear equations is an interesting and challenging problem. Marwah et al. (2023), examine nonlinear elliptic equations with linear growth conditions. Extending these results to parabolic equations would be an intriguing direction for future research.

- Many important equations in applied fields, such as finance or kinetic theory, are given in non-divergence form and often exhibit degeneracy. Investigating the curse of dimensionality in such equations could provide valuable insights. The regularity theory and Green's function estimates for these equations are discussed in (Di Francesco & Polidoro, 2006).

- In this work, we present a theoretical result using Green's function. Recent studies, such as (Boullé et al., 2022; Teng et al., 2022), demonstrate a growing interest in learning Green's functions by machine learning. Moreover, our results suggest that neural operators can be trained without encountering the curse of dimensionality since training Green's function can be applied to the faimily of PDEs instead of one instance of the equation. Exploring these areas further would be highly interesting.

- For connections between the Barron space defined here and the spectral Barron space, see Appendix G. Remarks on function spaces beyond Barron space and their relation to PDEs are in Appendix H. Numerical experiments validating our theorems are provided in Appendix O. We refer Appendix P for the estimate of $\Gamma(\frac{d+1}{2})/\Gamma(\frac{d}{2})$.

### ACKNOWLEDGMENTS

The authors thank the reviewers for their time and attention in reviewing this manuscript.

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

## A  NOTATIONS

**Points in Euclidean spaces**

| | |
|---|---|
| $\mathbb{R}$ | The set of real numbers |
| $\mathbb{R}^d$ | The $d$-dimensional Euclidean space, denoted by $\mathbb{R}^d$, is defined as the set of all $d$-tuples where each element of the tuple belongs to the real numbers |
| $\mathbb{R}^d_+$ | Half-space: $\{x \in \mathbb{R}^d : x_n > 0\}$, |
| $\mathbb{R}^d_-$ | $\{x \in \mathbb{R}^d : x_n < 0\}$, |
| $x, y, x_0$ | Points in $\mathbb{R}^d$ |
| $t, s, t_0$ | Points in $\mathbb{R}$ |
| $X, Y, X_0$ | Points in $\mathbb{R}^{d+1}$, where $X = (t, x), Y = (s, y), X_0 = (t_0, x_0)$ |
| $\lvert x - y \rvert$ | Euclidean distance between $x, y$ in $\mathbb{R}^d$, defined as $\lvert x - y \rvert = \sqrt{\sum_i^d (x_i - y_i)^2}$ |
| $\lvert X - Y \rvert$ | $\lvert X - Y \rvert = \max(\sqrt{\lvert t - s \rvert}, \lvert x - y \rvert)$ |
| $\langle x, y \rangle, \langle X, Y \rangle$ | inner product of two vectors $x, y \in \mathbb{R}^d$ and $X, Y \in \mathbb{R}^{d+1}$. |

**Sets in Euclidean spaces, calculus, matrix and indexing**

| | |
|---|---|
| $\mathscr{D}$ | A cylindrical domain $(0, t) \times \mathbb{R}^d \subset \mathbb{R}^{d+1}$, with $0 < t < \infty$ |
| $B_r(x_0)$ | $\{x \in \mathbb{R}^d : \lvert x - x_0 \rvert < r\}$ |
| $\partial_p \mathscr{D}$ | A parabolic boundary defined as $\partial_p \mathscr{D} = \{t = a\} \times \mathbb{R}^d$ |
| $\mathscr{U}$ | Subset of $\mathscr{U} \subset \mathbb{R}^{d+1}$. |
| $\Omega$ | open subset of $\mathbb{R}^d$ |
| $\partial \Omega$ | boundary of $\Omega$ |
| $\lvert \Omega \rvert, \lvert \mathscr{U} \rvert$ | volume of $\Omega$ and $\mathscr{U}$, respectively |
| $\nu$ | On the surface $\partial \Omega \subset \mathbb{R}^d$, $\nu = \nu^1, \cdots, \nu^d)$ denotes outward unit normal vector to the tangent plane of $\partial \Omega$ |
| $C_r(x_0)$ | $\{x \in \mathbb{R}^d : \max_{i=1,\cdots,d} \lvert x_i - x_{0,i} \rvert < r\}$ |
| $D_r(X_0)$ | $(t_0 - r^2, t_0 + r^2) \times C_r$ |
| $Q_r^-(X_0)$ | $(t_0 - r^2, t_0) \times B_r(x_0)$ |
| $Q_r^+(X_0)$ | $(t_0, t_0 + r^2) \times B_r(x_0)$ |
| $supp\, f$ | Support of $f$, i.e., $\{f(x) \neq 0\}$ |
| $D_i f$ | Partial derivative of $f$ with respect to $x_i$ |
| $\partial_t f$ | Partial derivative of $f$ with respect to the time variable |
| $Df$ | $(D_1 f, \cdots, D_d f)$ |
| $I_d$ | $d \times d$ identity matrix |
| $\mathbf{A}, \mathbf{b}, \mathbf{c},$ | Matrix and the vector coefficients of the operator $\mathcal{P}$ and $\mathcal{L}$ |
| $\lvert \mathbf{A} \rvert$ | For a $d \times d$ matrix, $\lvert A \rvert$ denotes determinant |
| $\mathbf{A}^\top$ | Matrix transpose |
| $\delta(x)$ | a Dirac delta function |
| $\Gamma(x)$ | Gamma function |

| | |
|---|---|
| $\displaystyle\int_{\Omega} f(x)\,dS$ | Integral over the domains $\Omega \subset \mathbb{R}^d$ |
| $\displaystyle\int_{\partial\Omega} f(x)\,dx$ | Surface integral over the boundary of the domains $\partial\Omega \subset \mathbb{R}^d$ |
| $\displaystyle\int_{\mathscr{U}} f(t,x)\,dx\,dt$ | Integral over the spatiotemporal domains $\mathscr{U} \subset \mathbb{R}^{d+1}$ |
| $(f)_{\mathscr{U}}, (f)_{\Omega}$ | Integral average of $f$ on the domain $\mathscr{U}$ and $\Omega$ respectively |
| $p'$ | Hölder conjugate, $\frac{1}{p'} + \frac{1}{p} = 1$ |

**Function spaces and related norm**

| | |
|---|---|
| $\|u\|_{L_p(\mathbb{R}^d)}$ | $\left(\int_{\mathbb{R}^d} |u(x)|^p\,dx\right)^{1/p}$ |
| $L_p(\mathbb{R}^d)$ | $\{u : \mathbb{R}^d \to \mathbb{R} : \|u\|_{L_p(\mathbb{R}^d)} < \infty\}$ |
| $W_p^1(\mathbb{R}^d)$ | $\{u : \mathbb{R}^d \to \mathbb{R} : \|u\|_{L_p(\mathbb{R}^d)}, \|Du\|_{L_p(\mathbb{R}^d)} < \infty\}$ |
| $C_c^k(\mathbb{R}^d)$ | set of all $k$ continuously differentiable functions with compact supports in $\mathbb{R}^d$ |
| $C_c^k(\Omega)$ | set of all $k$ continuously differentiable functions with compact supports in $\Omega$ |
| $\|u\|_{L_{p,q}(\mathscr{D})}$ | $\left(\int_0^b \left(\int_{\mathbb{R}^d} |u(t,x)|^p\,dx\right)^{q/p}\,dt\right)^{1/q}$ |
| $L_{p,q}(\mathscr{D})$ | $\{u : \mathscr{D} \to \mathbb{R} : \|u\|_{L_{p,q}(\mathscr{D})} < \infty\}$ |
| $L_p(\mathscr{D})$ | $L_{p,p}(\mathscr{D})$ |
| $W_2^{1,0}(\mathscr{D})$ | $\{u : \mathscr{D} \to \mathbb{R} : u, Du \in L_2(\mathscr{D})\}$ |
| $W_2^{1,1}(\mathscr{D})$ | $\{u : \mathscr{D} \to \mathbb{R} : u, Du, \partial_t u \in L_2(\mathscr{D})\}$ |
| $V_2(\mathscr{D})$ | $\{u : \mathscr{D} \to \mathbb{R} : u \in L_{2,\infty}(\mathscr{D}), Du \in L_2(\mathscr{D})\}$ |
| $\|u\|_{V_2(\mathscr{D})}$ | $\|Du\|_{L_2(\mathscr{D})} + \|u\|_{L_{2,\infty}(\mathscr{D})}$ |
| $\|u\|_{V_2^{1,0}(\mathscr{D})}$ | $\|Du\|_{L_2(\mathscr{D})} + \max_{a \le t \le b} \|u(t,\cdot)\|_{L_2(\mathbb{R}^d)}$ |
| $C_c^{k_1,k_2}(\mathscr{U})$ | set of all functions with $k_1$ continuously differentiable in $x$ variable and $k_2$ continuously differentiable in $t$ with compact supports in $\mathscr{U}$ |
| $\mathring{W}_2^{1,0}(\mathscr{D})$ | $u \in W_2^{1,0}(\mathscr{D})$ vanishes on $\partial_x \mathscr{D}$, equivalently, $u$ is the limit of functions from $C^{1,1}(\mathscr{D} \setminus \partial_x \mathscr{D})$ in $W_2^{1,0}(\mathscr{D})$ |
| $\mathring{V}_2(\mathscr{D})$ | $V_2(\mathscr{D}) \cap \mathring{W}_2^{1,0}(\mathscr{D})$ |
| $\mathcal{B}(\mathscr{D})$ | Barron space defined in Section 2.1. |

## B  PRECISE DEPENDENCY OF THE CONSTANTS

We gather the constants that appeared in this paper.

- $C$ is a universal constant depending only on $\lambda$ and $\Lambda$ but independent of the dimension $d \ge 2$, unless otherwise state. If it depends on other variables such as $\alpha$, then we denote $C(\alpha)$.
- $\beta(d,p)$: Embedding constants between spaces involving time are presented in Lemma 4, i.e.,

$$\beta(d,p) = \max\left(\frac{p(d-1)}{d}, 2\right)^{d\left(\frac{1}{2} - \frac{1}{p}\right)}.$$

- $C_{JN}(p,d)$. Constants in John-Nirenberg inequality presented in Lemma 3,

- $C_{energy}$ Constants for the energy inequailty; see equation 21.

## C  ADDITIONL DETAILS ON REGULARITY THEORY

This section provides additional theoretical details on regularity theory, such as BMO space and VMO space.

**Definition 3** (BMO($\mathbb{R}^d$) space)**.** *A locally integrable function $f$ on $\mathbb{R}^d$ is said to be in the space of bounded mean oscillation or $f \in \mathrm{BMO}(\mathbb{R}^d)$ if there exists a constant $C > 0$ such that*

$$\|f\|_{\mathrm{BMO}} := \sup_{K \subset \mathbb{R}^d} \left( \frac{1}{|K|} \int_K |f(x) - (f)_K|^2 \, dx \right) \leq C,$$

*where*

$$(f)_K = \frac{1}{|K|} \int_K f(x) \quad and \quad |K| = volume \ of \ the \ cube \ K.$$

**Definition 4** (VMO($\mathbb{R}^d$) space)**.** *A locally integrable function $f$ on $\mathbb{R}^d$ is said to be in the space of vanishing mean oscillation or $f \in \mathrm{VMO}(\mathbb{R}^d)$ if we have*

$$\lim_{|K| \to 0} \sup_{K \subset \mathbb{R}^d} \left( \frac{1}{|K|} \int_K |f(x) - (f)_K|^2 \, dx \right) = 0$$

**Remark 5.** *It is worth mentioning that the definition of BMO and VMO can be changed to the ball instead of the cube; see, for instance, (Stein, 1993). Also, as a result of the John-Nirenberg inequality, the integral average*

$$\int_K |f(x) - (f)_K|^2 \, dx \, dt$$

*can be replaced to*

$$\int_K |f(x) - (f)_K|^p \, dx \, dt$$

*for $1 \leq p < \infty$, in our assumptions on the coefficients. The precise statement of the John-Nirenberg inequality is stated below.*

**Lemma 3** (John-Nirenberg lemma)**.** *For all $f \in BMO(\mathbb{R}^d)$ and for all $0 < p < \infty$, there exists a finite constant $C_{JN}(p, n) > 0$ such that*

$$\sup_{K \subset \mathbb{R}^d} \left( \frac{1}{|K|} \int_K |f(x) - (f)_K|^p \, dx \right)^{\frac{1}{p}} \leq C_{JN}(p, n) \|f\|_{\mathrm{BMO}(\mathbb{R}^d)}.$$

$$C_{JN}(p, d) = (p\Gamma(p))^{\frac{1}{p}} \, e^{\frac{1}{p}+1} 2^d. \tag{15}$$

For the proof, we refer (Grafakos, 2009, Corollary 7.1.8).

## D  REMARKS ON VMO($\mathbb{R}^d$) SPACE

In this section, we provide examples of functions that belong to and do not belong to the VMO space. For more nformation regarding VMO space, we refer to (Brezis & Nirenberg, 1995; Grafakos, 2009; Duoandikoetxea, 2024).

- Examples of functions $f \in$ VMO spaces:
    - $f(x)$ is an uniformly continuous funciton.
    - $f(x)$ is a continuous function with a compact support.
    - If $f(x) \in$ VMO and $L(x)$ is an Lipschitz funciton, then $L(f) \in$ VMO.
    - $f(x) \in W_d^1$ then $f \in$ VMO.
    - $f(x) \in W_p^s$ with $sp = d$, then $f \in$ VMO.
    - $f(x) = \log|\log(|x|)|$.

– $f(x) = |\log(|x|)|^\alpha$ for $0 < \alpha < 1$.
– $f(x) = \sin(\log(|\log(|x|)))$.
– $f(x) = |x|\sin(1/|x|)$.

- Examples of functions $f \notin$ VMO spaces:

    – $f(x) = \log(|x|)$.
    – $f(x) = |\log(|x|)|^p$ with $1 < p < \infty$.
    – $f(x) = \chi_{[0,1]^d}(x)$, characteristic function on $[0,1]^d$.
    – $f(x) = \sin(1/|x|)$.
    – $f(x) = \sin(\log(|x|))$
    – $f(x) = \sin(1/|x|^2)$

For detailed definition on fractioanl Sobolev space, $W_p^s$, we refer to (Di Nezza et al., 2012).

## E   REMARKS ON THE CONTINUITY OF THE WEAK SOLUTIONS

Continuity of the solutions to equation 2 and equation 5 has been widely studied since the De Giorgi-Nash-Moser theorem. We briefly provide some conditions under which the solutions belong to the space of continuous functions.

For the elliptic case, by (Gilbarg et al., 1977, Theorem 8.24), if $f \in L_{q,\mathrm{loc}}(\mathbb{R}^d)$ with $q > \frac{d}{2}$, then the weak solution of equation 5 belongs to the space of Hölder continuous functions. For the parabolic case, by (Aronson & Serrin, 1967, Theorem 4), the weak solution of equation 20 is Hölder continuous for $t > 0$ if $f \in L_{q,\mathrm{loc}}(\mathbb{R}^d)$ with $q > \frac{d+2}{2}$. Furthermore, by (Kim & Xu, 2021, Theorem 3.1), the operator equation 1 has a Green's function $G$, and the weak solution of equation 2 can be represented as in equation 8. Thus, $u(t,x) \to g(x)$ as $t \to 0$. Therefore, if the initial function $g(x)$ is continuous, then $u(t,x)$, the weak solution of equation 2, is continuous.

## F   REMARKS ON ACTIVATION AND BARRON SPACE

In this work, we choose the activation function as $\mathrm{ReLU}(x)$. For a non-ReLU activation function, the Barron norm is defined as

$$\|f\|_{\mathcal{B}(\Omega)} := \inf \left\{ \int_{\mathbb{R}^{d+2}} |a|(\|w\| + |b| + 1)\,\pi(da \otimes dw \otimes db) : \pi \in A_f(\Omega) \right\}.$$

Compared with equation 6, for a Barron function with a non-ReLU activation function, there is an additional $+1$ term. A direct approximation theorem, such as Property 1, holds if

$$\int_{\mathbb{R}^d} |\sigma''(x)|(|x| + 1)\,dx < \infty.$$

However, to the best of the authors' knowledge, an inverse approximation result, such as Proposition 2, is not currently known. Addressing these technical difficulties remains a topic for future research.

## G   CONNECTION WITH SPECTRAL BARRON SPACES

The Barron spaces defined in Section 2.1 and the spectral Barron spaces are closely connected. For a compact set $\Omega \subset \mathbb{R}^d$ and a function $f : \Omega \to \mathbb{R}$, the spectral Barron norm is defined as:

$$\|f\|_{\mathcal{F}_s(\Omega)} = \inf_{f_e|_\Omega = f} \int_{\mathbb{R}^d} (1 + \|\xi\|_\Omega)^s |\hat{f}_e(\xi)|\,d\xi, \tag{16}$$

where $f_e$ is an extension of $f$ to $\mathbb{R}^d$, and $\|\xi\|_\Omega = \sup_{x \in \Omega} |\xi^T x|$. By (Wu, 2023, Theorem 1.4), we have:

$$\frac{\varepsilon}{C}\|f\|_{\mathcal{F}_{1-\varepsilon}(\Omega)} \le \|f\|_{\mathcal{B}(\Omega)} \le C\|f\|_{\mathcal{F}_2(\Omega)} \quad \forall \varepsilon \in (0,1).$$

Thus, our results can be directly applied to the Barron space with the Barron norm given in equation 16.

On the other hand, if we define a Barron space for functions over the entire domain, as considered in (Chen et al., 2023), the spectral Barron space is characterized by the Barron norm:

$$\|f\|_{\mathcal{F}_s(\mathbb{R}^d)} := \int_{\mathbb{R}^d} (1 + \|\xi\|^2)^{\frac{s}{2}} |\hat{f}(\xi)| \, d\xi. \tag{17}$$

In this setting, a significant technical challenge arises due to the absence of an inverse approximation result, as presented in Proposition 2. Addressing this issue and studying regularity results within the spectral Barron space is an intriguing direction for future research.

## H  REMARKS ON FUNCTION SPACES BEYOND BARRON SPACES AND MULTI-LAYERS NEURAL NETWORKS

In practical applications, two-layer networks are rarely employed, making the theoretical analysis of function approximation using multi-layer networks an increasingly compelling topic. In (Bresler & Nagaraj, 2020), the authors demonstrate that if $u \in \mathcal{F}_{1/K}(\mathbb{R}^d)$, there exists a $K$-neuron ReLU$(x)$ network $u_\theta$ capable of approximating $u$. However, as explained in Section G, our results cannot be directly extended to $u \in \mathcal{F}_{1/K}(\mathbb{R}^d)$ due to the absence of inverse approximation results for the spectral Barron spaces.

The study of function spaces for multi-layer networks remains highly underexplored. Beyond the limited existing works, such as (Wojtowytsch et al., 2020; Chen, 2023; 2024), there has been little research into defining meaningful functional properties of deep neural networks. Expanding these investigations is crucial for advancing our understanding of multi-layer networks, particularly to leverage such function spaces for approximating solutions to partial differential equations (PDEs).

While shallow networks have been analyzed extensively, deep networks are still at a nascent stage in terms of both their theoretical function spaces and practical applications to PDE approximation. Further exploration of these areas is a topic for future research.

## I  PRELIMINARIES LEMMAS

### EMBEDDING INEQUALITIES

Embedding inequalities play a crucial role in having energy estimates for the solution of elliptic and parabolic equations. The proof of the following well-known embedding inequality can be found in (Ladyženskaja, 1968, pp. 74-75) and the precise dependency of $\beta$ can be found in (Ladyženskaja, 1968, Chapter II, Theorem 2.2).

**Lemma 4.** *Let $d \geq 1$ be an integer, and consider exponents $\tilde{p}$ and $\tilde{q}$ that satisfy the relation*

$$\frac{d}{\tilde{p}} + \frac{2}{\tilde{q}} = \frac{d}{2}.$$

*The range of these exponents satisfy*

$$\begin{cases} \tilde{p} \in \left[2, \dfrac{2d}{d-2}\right], & \tilde{q} \in [2, \infty] & \text{if } d \geq 3, \\ \tilde{p} \in [2, \infty), & \tilde{q} \in (2, \infty] & \text{if } d = 2. \end{cases} \tag{18}$$

*Then, the following inequality holds*

$$\|u\|_{L_{\tilde{p}, \tilde{q}}(\mathscr{D})} \leq \beta(d, \tilde{p}) \|u\|_{V_2(\mathscr{D})} \quad \forall u \in \overset{\circ}{V}_2(\mathscr{D}) \tag{19}$$

*with*

$$\beta(d, \tilde{p}) = \max\left(\frac{\tilde{p}(d-1)}{d}, 2\right)^{d\left(\frac{1}{2} - \frac{1}{\tilde{p}}\right)}.$$

We present similar embedding results for the function space without the time variable, whose proof can be found in (Talenti, 1976), where the bound is presented using the Gamma function in the original text. In this work, we use Stirling's approximation to reduce dimensionality as much as possible.

**Lemma 5.** *For all $u \in W_2^1(\mathbb{R}^d)$, we have*

$$\left( \int_{\mathbb{R}^d} |u|^{2^*} \, dx \right)^{1/(2^*)} \leq \gamma(d) \left( \int_{\mathbb{R}^d} |Du|^2 \, dx \right)^{1/2},$$

*where $2^* = 2d/(d-2)$ and $\gamma(d)$ is defined as*

$$\gamma(d) = \frac{4}{\sqrt{\pi d}} \left( \frac{1}{d-2} \right)^{1-\frac{1}{2}}$$

EXISTENCE AND UNIQUENESS OF WEAK SOLUTIONS

**Lemma 6.** *Suppose that $f \in L_{(2n+4)/(n+4)}(\mathscr{D})$ and that $g \in L_2(\mathbb{R}^d)$. Then equation 2 has a unique weak solution $u \in \mathring{V}_2(\mathscr{D})$ with the following energy estimates*

$$\|u\|_{V_2(\mathscr{D})} \leq C_{Energy} \left( \|g\|_{L_2(\mathbb{R}^d)} + \|f\|_{L_{\frac{2d+4}{d+4}}(\mathscr{D})} \right), \tag{20}$$

*where*

$$C_{energy} = \left( 2 + \lambda^{-1} + \beta \left( d, \frac{2d+4}{d} \right) \right). \tag{21}$$

*Proof.* We take $u(t, x)$ as a test function in equation 3 to find for all $t_1 \in [1, b]$

$$\int_{\mathbb{R}^d} u^2(t_1, x) \, dx - \frac{1}{2} \int_0^{t_1} \frac{d}{dt} \left( \int_{\mathbb{R}^d} u^2 \, dx \right) dt + \int_0^{t_1} \int_{\mathbb{R}^d} (a^{ij} D_j u + b^i u) D_i u \, dx \, dt$$

$$+ \int_0^{t_1} \int_{\mathbb{R}^d} c^i u D_i u + d(x, t) u^2 \, dx \, dt - \int_0^{t_1} \int_{\mathbb{R}^d} f u \, dx \, dt$$

$$= \frac{1}{2} \int_{\mathbb{R}^d} u^2(t_1, x) \, dx + \frac{1}{2} \int_{\mathbb{R}^d} g^2 \, dx + \int_0^{t_1} \int_{\mathbb{R}^d} (a^{ij} D_j u + b^i u) D_i u \, dx \, dt$$

$$+ \int_0^{t_1} \int_{\mathbb{R}^d} c^i u D_i u + d(x, t) u^2 \, dx \, dt - \int_0^{t_1} \int_{\mathbb{R}^d} f u \, dx \, dt$$

$$= \int_{\mathbb{R}^d} g^2(x) \, dx$$

Note that since $u$ does not have a regularity in time direction, we take a Steklov averge $u_h$ as a test funciton in equation 3 and then take $h \to 0$ as described in (Ladyženskaja, 1968, Chapter III.2) or (DiBenedetto, 2012, Chapter I.3). Rewriting equality above, we have

$$\frac{1}{2} \int_{\mathbb{R}^d} u^2(t_1, x) \, dx + \int_0^{t_1} \int_{\mathbb{R}^d} (a^{ij} D_j u + b^i u) D_i u \, dx \, dt + \int_0^{t_1} \int_{\mathbb{R}^d} c^i u D_i u + d(x, t) u^2 \, dx \, dt$$

$$= \frac{1}{2} \int_{\mathbb{R}^d} g^2(x) \, dx + \int_0^{t_1} \int_{\mathbb{R}^d} f u \, dx \, dt. \tag{22}$$

By the Assumption 1 - (3) and the fact that $u^2 \geq 0$, we have

$$\int_0^{t_1} \int_{\mathbb{R}^d} d(x, t) u^2 + 2 b^i u D_i u \geq 0 \quad \text{and} \quad -\int_0^{t_1} \int_{\mathbb{R}^d} (b^i - c^i) u D_i u \geq 0.$$

This inequalities, equation 23 and Assumption 1 - (1), we have

$$\frac{1}{2} \int_{\mathbb{R}^d} u^2(t_1, x) \, dx + \lambda \int_0^{t_1} \int_{\mathbb{R}^d} |Du|^2 \, dx \, dt$$

$$\leq \frac{1}{2} \int_{\mathbb{R}^d} g^2(x) \, dx + \int_0^{t_1} \int_{\mathbb{R}^d} f u \, dx \, dt. \tag{23}$$

By Hölder's inequality and equation 19, we have

$$\int_0^{t_1} \int_{\mathbb{R}^d} fu \, dx \, dt \le \|f\|_{L_{(2d+4)/(d+4)}(\mathscr{D})} \|u\|_{L_{(2d+4)/d}(\mathscr{D})} \tag{24}$$

$$\le \beta(d, (2d+4)/d) \|f\|_{L_{(2d+4)/(d+4)}(\mathscr{D})} \|u\|_{V_2(\mathscr{D})}.$$

By equation 23 and equation 24, we have equation 20 by direct computation. The existence and uniqueness can be derived from the energy estimate equation 20 via Galekin's method as in (Ladyženskaja, 1968, Chapter III.4). $\qquad\square$

For the elliptic case, we have a similar result.

**Lemma 7.** *Suppose that $f \in L_{(d+2)/(2d)}(\mathbb{R}^d)$. Then equation 5 has the unique weak solution $u \in W_2^1(\mathbb{R}^d)$ with the following energy estimates*

$$\|u\|_{W_2^1(\mathbb{R}^d)} \le C(\gamma(d), \lambda, d_{min}) \|f\|_{L_{\frac{d+2}{2d}}(\mathbb{R}^d)}.$$

*for some constant $C(\gamma(d), \lambda, d_{min}) > 0$.*

The process of the proof is similar to that of Lemma 6. By testing $u$ in equation 4 and using Lemma 7 along with Young's inequality, the result follows.

## J    PROOF OF LEMMA 2

*Proof.* We are intend to show that equation 3 holds for all test function $\phi \in C_c^{1,1}(\mathscr{D})$. Since $f$ and $g$ are continuously differentiable, $u$ is continuously differentiable. Moreover for all $(t, x) \in \mathscr{D} \setminus \mathcal{N}$, we have

$$\mathcal{P}_\delta u = f$$

and $u(0, x) = g(x)$ holds assuming that $(0, x) \notin \mathcal{N}$. We arbitrary choose $\phi \in C_c^{1,1}(\mathscr{D})$ and let $\psi^k(x) \in C_c^{1,1}(D_{R_\delta}^k)$ satisfying $|\psi^k| \le 1$ and let us denote

$$\phi_1^k = \phi \psi^k, \quad \phi_2^k = \phi(1 - \psi_2^k) \chi_{D_{R_\delta}^k} \quad \text{and} \quad \psi = \sum_{k : D_{R_\delta}^k \cap supp \, \phi \neq \emptyset} \psi^k.$$

Since $supp \, \phi$ is compact, there exist finite numbers of such $k$ satisfying $D_{R_\delta}^k \cap supp \, \phi \neq \emptyset$ and the summation converges. We may further assume that for some $\varepsilon > 0$, $supp \, \psi^k \subset D_{R_\delta - \varepsilon}^k$, where the center is the same as $D_{R_\delta}^k$ for each $k$. Therefore, the choice of $\psi^k$ depends on $\varepsilon > 0$. Then, we decompose $\phi$ as

$$\phi = \psi \phi + (1 - \psi)\phi =: \phi_1 + \phi_2.$$

Since each $\phi^k$ has a support in $D_{R_\delta}^k$, for almost all $t_1 \in (0, t)$, we have

$$I_\delta(t_1, u_\delta, \phi_1) = \int_{\mathbb{R}^d} g(x)\phi_1(0, x) \, dx,$$

where $I_\delta(t, u, \phi)$ represent linear operator defined as in equation 3 replacing $a^{ij}(t, x), b^i(t, x), c^i(t, x), d(t, x)$ by $a_\delta^{ij}(t, x), b_\delta^i(t, x), c_\delta^i(t, x), d_\delta(t, x)$. It remains to show that taking $\varepsilon \to 0$, we have

$$I(t_1, u_\delta, \phi_2) - \int_{\mathbb{R}^d} g(x)\phi_2(0, x) \, dx \to 0,$$

since the value

$$I(t_1, u_\delta, \phi) - \int_{\mathbb{R}^d} g(x)\phi \, dx = I(t_1, u_\delta, \phi_1) - \int_{\mathbb{R}^d} g(x)\phi_1 \, dx + I(t_1, u_\delta, \phi_2) - \int_{\mathbb{R}^d} g(x)\phi_2 \, dx$$

$$= I(t_1, u_\delta, \phi_2) - \int_{\mathbb{R}^d} g(x)\phi_2 \, dx$$

does not depends on $\varepsilon > 0$, this implies that

$$I(t_1, u_\delta, \phi) - \int_{\mathbb{R}^d} g(x)\phi \, dx = 0.$$

Let us estimate each term. We have that

$$I(t_1, u_\delta, \phi_2) - \int_{\mathbb{R}^d} g(x)\phi_2\, dx$$

$$= \int_{\mathbb{R}^d} u_\delta(t_1, x)\phi_2(t_1, x)\, dx - \int_0^{t_1} \int_{\mathbb{R}^d} u_\delta \partial_t \phi_2\, dx\, dt$$

$$+ \int_0^{t_1} \int_{\mathbb{R}^d} (a_\delta^{ij} D_j u_\delta + b_\delta^i u_\delta) D_i \phi_2\, dx\, dt + \int_0^{t_1} \int_{\mathbb{R}^d} (c_\delta^i D_i u_\delta + d_\delta u_\delta)\phi_2\, dx\, dt$$

$$- \int_0^{t_1} \int_{\mathbb{R}^d} f\phi_2\, dx\, dt - \int_{\mathbb{R}^d} g(x)\phi_2\, dx$$

$$=: I_1 + I_2 + I_3 + I_4 + I_5 + I_6.$$

Note that as $supp\, \phi_2 \to 0$ as $\varepsilon \to 0$, we have $I_1, I_4, I_5, I_6 \to 0$. Therefore, we only need to consider $I_2$ and $I_3$.

We shall use the fact that since $f \in C_2^1(\mathscr{D})$ and $g \in C_1(\mathbb{R}^d)$, $u_\delta$ has a differentiability in $x$. For the $t$ variable, we use the decomposition technique. That is, decompose the domain and then by the parts as in below

$$I_2 = \int_0^{t_1} \int_{\mathbb{R}^d} \partial_t u_\delta \phi_2 dx\, dt$$

$$= \sum_k \int_{t_k - (R_\delta^k)^2}^{t_k + (R_\delta^k)^2} \int_{C_{R_\delta}^k} \partial_t u_\delta \phi_2 dx\, dt$$

$$= \sum_k \int_{C_{R_\delta}^k} u_\delta(t_k + (R_\delta^k)^2, x)\phi_2(t_k + (R_\delta^k)^2, x)\, dx$$

$$- \int_{C_{R_\delta}^k} u_\delta(t_k - (R_\delta^k)^2, x)\phi_2(t_k - (R_\delta^k)^2, x)\, dx$$

$$- \sum_k \int_0^{t_1} \int_{{C_{R_\delta}^k}} \partial_t u_\delta \phi_2 dx\, dt.$$

Since $\partial_t u_\delta(x, t)$ is well-defined inside each region $D_{R_\delta}^k$ and belongs to $L^1(\mathscr{D})$ and , $supp\, \phi_2 \to 0$ implies that $I_2 \to 0$. Next, we have

$$I_3 = \sum_{k: D_{R_\delta}^k \cap supp\, \phi \neq \emptyset} \int_0^{t_1} \int_{D_{R_\delta}^k} ((a^{ij})_{D_{R_\delta}^k} D_j u_\delta + ((b^i)_{D_{R_\delta}^k} u_\delta) D_i \phi_2^k\, dx\, dt$$

$$\sum_{k: D_{R_\delta}^k \cap supp\, \phi \neq \emptyset} ((a^{ij})_{D_{R_\delta}^k} \int_0^{t_1} \int_{D_{R_\delta}^k} D_j u_\delta D_i \phi_2^k\, dx\, dt + +(b^i)_{D_{R_\delta}^k} \int_0^{t_1} \int_{D_{R_\delta}^k} u_\delta D_i \phi_2^k\, dx\, dt.$$

As mentioned earlier, as $D_{R_\delta}^k \cap supp\, \phi \neq \emptyset$ only has finite $k$, we only need to show that

$$\int_0^{t_1} \int_{D_{R_\delta}^k} D_j u_\delta D_i \phi_2^k\, dx\, dt \to 0 \quad \text{and} \quad \int_0^{t_1} \int_{D_{R_\delta}^k} u_\delta D_i \phi_2^k\, dx\, dt \to 0.$$

Through the integration of parts, we have

$$\int_0^{t_1} \int_{D_{R_\delta}^k} D_j u_\delta D_i \phi_2^k\, dx\, dt$$

$$= \int_0^{t_1} \int_{D_{R_\delta}^k} D_{ij} u_\delta \phi_2^k\, dx\, dt - \int_0^{t_1} \int_{\partial(D_{R_\delta}^k - D_{R_\delta - \varepsilon}^k)} D_{ij} u_\delta \phi_2^k \nu^i\, dS(x)\, dt.$$

Here, for $A \subset \mathbb{R}^d$, $\partial A$ implies the boundary of $A$, $\nu^i$ denotes the outward unit average vector to the integral domain, and $dS$ implies integral on the boundary. First term converges to $0$ as $\varepsilon \to 0$ as

$u \in W_2^1(\mathscr{D})$ and $|supp \, \phi_2^k| \to 0$. For the second term, we use the Trace inequality, (Evans, 2022, Chapter 5.5, Theorem 1), to find

$$\int_0^{t_1} \int_{\partial(D_{R_\delta}^k - D_{R_\delta}^k)} |D_i u_\delta \phi_2^k \nu^i| \, dS(x) \, dt \le C(d) \|\phi_2^k\|_{L_\infty(\mathscr{D})} \int_0^{t_1} \int_{D_{R_\delta}^k - D_{R_\delta}^k} |D^2 u_\delta| \, dx.$$

This also goes to 0 as $|D_{R_\delta}^k - D_{R_\delta - \varepsilon}^k| \to 0$ as well. Finally, repeating the same procedure (integration by parts, using the fact that $u_\delta$ has sufficient regularity, trace inequality), we have

$$\int_0^{t_1} \int_{D_{R_\delta}^k} u_\delta D_i \phi_2^k \, dx \, dt \to 0,$$

as $\varepsilon \to 0$, which completes the proof.

$\square$

## K  PROOF OF THEOREM 3

The argument in this proof is motivated by the one presented in (Weinan & Wojtowytsch, 2022, Lemma 3). We extend these results to a more general equation.

*Proof.* For the simplicity of the notation, we shall remove the points $(t, x)$ and $(s, y)$ for the coefficients. We remark that the coefficients are piecewise constants functions. From the definition, we have

$u_{\text{hom}}(t, x)$

$$= \int_{\mathbb{R}^d} \frac{g(y)}{(4\pi t)^{d/2} |\det(\mathbf{A}_\delta)|^{1/2}} \exp\left\{ -\frac{\langle \mathbf{A}_\delta^{-1}(y - x - (\mathbf{c}_\delta - \mathbf{b}_\delta)t), y - x - (\mathbf{c}_\delta - \mathbf{b}_\delta)t \rangle}{4t} - d_\delta t \right\} dy.$$

Since $A_\delta$ is a symmetric metric, there exists $A_\delta = Q_\delta^\top D_\delta^{\frac{1}{2}} D_\delta^{\frac{1}{2}} Q_\delta$, where $D_\delta^{\frac{1}{2}}$ is a symmetric metric where diagonal elements are square root of the eigenvalues and $Q_\delta$ is an orthogonal matrix satisfying

$$Q_\delta^\top Q_\delta = Q_\delta Q_\delta^\top = I.$$

Then, we have

$$\frac{\langle \mathbf{A}_\delta^{-1}(y - x - (\mathbf{c}_\delta - \mathbf{b}_\delta)t), y - x - (\mathbf{c}_\delta - \mathbf{b}_\delta)t \rangle}{4t}$$
$$= \frac{\langle D^{-\frac{1}{2}} Q_\delta(y - x - (\mathbf{c}_\delta - \mathbf{b}_\delta)t), D^{-\frac{1}{2}} Q_\delta(y - x - (\mathbf{c}_\delta - \mathbf{b}_\delta)t) \rangle}{4t}.$$

Substituting $y$ with

$$z = D^{-\frac{1}{2}} Q_\delta \frac{y - x - (\mathbf{c}_\delta - \mathbf{b}_\delta)t}{\sqrt{t}}$$

we find that

$u_{\text{hom}}(t, x)$

$$= \int_{\mathbb{R}^d} \frac{g(x + (\mathbf{c}_\delta - \mathbf{b}_\delta)t + \sqrt{t} Q_\delta^\top D_\delta^{\frac{1}{2}} z)}{(4\pi t)^{d/2}} \exp\left\{ -\frac{|z|^2}{4t} - d_\delta t \right\} dz$$

$$= e^{-d_\delta t} \int_{\mathbb{R}^d} \int_{\mathbb{R}^{d+2}} \frac{a\sigma(w^\top(x + (\mathbf{c}_\delta - \mathbf{b}_\delta)t + \sqrt{t} Q_\delta^\top D_\delta^{\frac{1}{2}} z) + b)}{(4\pi t)^{d/2}} d\pi(a, w, b) e^{\left(-\frac{|z|^2}{4t}\right)} dz$$

$$= e^{-d_\delta t} \int_{\mathbb{R}^{d+2}} \int_{\mathbb{R}^d} \frac{a\sigma(w^\top(x + (\mathbf{c}_\delta - \mathbf{b}_\delta)t + \sqrt{t} Q_\delta^\top D_\delta^{\frac{1}{2}} z) + b)}{(4\pi t)^{d/2}} e^{\left(-\frac{|z|^2}{4t}\right)} dz d\pi(a, w, b).$$

Using the fact that $d_\delta \ge 0$ and the definition of the Barron space, we find

$$\|u_{\delta,\text{hom}}(t, \cdot)\|_{\mathcal{B}(\mathbb{R}^d)} \le \int_{\mathbb{R}^{d+2}} |a||w| + |a||w||c_\delta - b_\delta|t + |a||w^\top Q_\delta^\top D_\delta^{\frac{1}{2}}|\sqrt{t}) d\pi(a, w, b)$$

$$\le (1 + \Lambda^{\frac{1}{2}} \sqrt{t} + \|c - b\|_{L^\infty(\mathscr{D})} t) \|g\|_{\mathcal{B}(\mathbb{R}^d)}.$$

Next, we estimate the Barron norm of $u_{\delta,\text{inhom}}$. Using the previous result, the fact that

$$\|f(t,\cdot)\|_{\mathcal{B}(\mathbb{R}^d)} \leq \max(t,1)\|f(t,\cdot)\|_{\mathcal{B}((0,t)\times\mathbb{R}^d)},$$

and the representation formula equation 10, we find

$$
\begin{aligned}
&\|u_{\delta,\text{inhom}}(t,\cdot)\|_{B(\mathbb{R}^d)} \\
&\leq \int_0^t \|f(s,\cdot)\|_{B(\mathbb{R}^d)}(1 + \Lambda^{\frac{1}{2}}\sqrt{t-s} + \|c-b\|_{L^\infty(\mathscr{D})}(t-s))\,ds \\
&\leq \|f\|_{B((0,t)\times\mathbb{R}^d)} \int_0^t \max(1,s)(1 + \Lambda^{\frac{1}{2}}\sqrt{t-s} + \|c-b\|_{L^\infty(\mathscr{D})}(t-s))\,ds \\
&\leq \|f\|_{B((0,t)\times\mathbb{R}^d)} \left( t + \frac{t^2}{2} + \Lambda^{\frac{1}{2}}\left(\frac{2t^{\frac{3}{2}}}{3} + \frac{2t^{\frac{5}{2}}}{5}\right) + \|c-b\|_{L^\infty(\mathscr{D})}\left(\frac{t^2}{2} + \frac{t^3}{6}\right)\right).
\end{aligned}
$$

$\square$

## L  PROOF OF THEOREM 4

*Proof.* Note that we have

$$
\begin{cases}
\mathcal{P}_\delta(u - u_\delta) = (P - P_\delta)u_\delta & \text{in } \mathscr{D}, \\
u(0,\cdot) - u_\delta(0,\cdot) = 0 & \text{on } \mathbb{R}^d.
\end{cases}
\tag{25}
$$

Let $\xi(x)$ be a smooth cut-off function satisfying $\xi = 1$ on $B_R$ and $\xi \in C_c^1(B_{2R})$ with

$$|D\xi| \leq \frac{2}{R}.$$

For simplicity of the notation, we denote $\tilde{C}_1$ for the constant depends on $t, d, \lambda, \Lambda, \|f\|_{L_{\frac{2d+4}{d+4}}(\mathscr{D})}, \|g\|_{L_2(\mathbb{R}^d)}$. The constant $\tilde{C}_2$ depends on also on $R$ as well. That is, $\tilde{C}_1$ is independent of $R > 0$. Also, it is observed that for $p \geq 2$

$$\int_0^{t_1}\int_{B_R} |a^{ij} - a_\delta^{ij}|^p + |b^i - b_\delta^i|^p + |c^i - c_\delta^i|^p + |d(x,t) - d_\delta|^p \, dx\,dt \leq \tilde{C}_2(p)\delta.$$

Here, $\tilde{C}_2(p)$ implies additional depenedecy on $p \geq 2$. We test both sides by $\xi^2(u - u_\delta)$ on both sides. Then, on the left-hand side, we use the Assumption 1 - (3) and the Assumption 1 - (1), as in the procedure provided in Lemam 6, to have

$$
\begin{aligned}
&\frac{1}{2}\int_{\mathbb{R}^d} \xi^2(u(t_1,x) - u_\delta(t_1,x))^2\,dx + \frac{\lambda}{2}\int_0^{t_1}\int_{\mathbb{R}^d} \xi^2|Du - Du_\delta|^2\,dx\,dt \\
&\leq LHS + \frac{\Lambda^2}{2\lambda}\int_0^{t_1}\int_{\mathbb{R}^d} |D\xi|^2|u - u_\delta|^2\,dx\,dt \\
&\leq LHS + \frac{\tilde{C}_1}{R^2},
\end{aligned}
$$

where $LHS$ denotes left hand side of the equation after testing $\xi^2(u - u_\delta)$ to equation 25. The right-hand side can be estimated as below:

$$RHS \leq \int_0^{t_1} \int_{\mathbb{R}^d} (|a^{ij} - a_\delta^{ij}||D_j u_\delta| + |b^i - b_\delta^i||u_\delta|)|D_i(\xi^2(u - u_\delta))| \, dx \, dt$$

$$+ \int_0^{t_1} \int_{\mathbb{R}^d} \xi^2(|c^i - c_\delta^i||D_i u_\delta| + |d(x,t) - d_\delta||u_\delta|)|u - u_\delta| \, dx \, dt$$

$$\leq \int_0^{t_1} \int_{B_{2R}} \xi^2|a^{ij} - a_\delta^{ij}||D_j u_\delta||D_i u - D_i u_\delta| \, dx \, dt$$

$$+ 2 \int_0^{t_1} \int_{B_{2R}} \xi|a^{ij} - a_\delta^{ij}||D_j u_\delta||u - u_\delta||D_i\xi| \, dx \, dt$$

$$+ \int_0^{t_1} \int_{B_{2R}} \xi^2|b^i - b_\delta^i||u_\delta||D_i u - D_i u_\delta| \, dx \, dt$$

$$+ 2 \int_0^{t_1} \int_{B_{2R}} \xi|b^i - b_\delta^i||u_\delta||u - u_\delta||D_i\xi| \, dx \, dt$$

$$+ \int_0^{t_1} \int_{B_{2R}} \xi^2|c^i - c_\delta^i||D_i u_\delta||u - u_\delta| \, dx \, dt$$

$$+ \int_0^{t_1} \int_{B_{2R}} \xi^2|d(x,t) - d_\delta||u_\delta||u - u_\delta| \, dx \, dt$$

$$=: I_1 + I_2 + I_3 + I_4 + I_5 + I_6.$$

Let us estimate each term. Using the Young's inequality with Hölder's inequality for some $p > 1$, we find that

$$I_1 \leq \frac{\lambda}{4} \int_0^{t_1} \int_{B_{2R}} \xi^2|D_i u - D_i u_\delta|^2 \, dx \, dt$$

$$+ \left( \int_0^{t_1} \int_{B_{2R}} |a^{ij} - a_\delta^{ij}|^{\frac{2p}{p-1}} \, dx \, dt \right)^{\frac{p-1}{p}} \left( \int_0^{t_1} \int_{B_{2R}} |Du_\delta|^{2p} \, dx \, dt \right)^{\frac{1}{p}}.$$

By (Giaquinta & Struwe, 1982, Theorem 2.1) and there exists $p = p(d, \lambda, \Lambda) > 1$ satisfying

$$\left( \frac{1}{t_1|B_2R|} \int_0^{t_1} \int_{B_{2R}} |Du_\delta|^{2p} \, dx \, dt \right)^{\frac{1}{p}} \leq C(d, \lambda, \Lambda) \frac{1}{t_1|B_2R|} \int_0^{t_1} \int_{B_{2R}} |Du_\delta|^2 \, dx \, dt.$$

From this inequality and Assumption 1 - (2), we have

$$I_1 \leq \frac{\lambda}{4} \int_0^{t_1} \int_{B_{2R}} \xi^2|D_i u - D_i u_\delta|^2 \, dx \, dt + \tilde{C}_2 \delta^{\frac{p-1}{p}} \int_0^{t_1} \int_{\mathcal{D}} |Du_\delta|^2 \, dx \, dt.$$

By the Hölder's inequality, Lemma 4 and Lemm 6, we find

$$I_2 \leq \frac{2}{R} \left( \int_0^{t_1} \int_{B_{2R}} |\mathbf{A} - \mathbf{A}_\delta|^{\frac{d+4}{2}} \, dx \, dt \right)^{\frac{2}{2d+4}} \left( \int_0^{t_1} \int_{B_{2R}} |Du_\delta|^2 \, dx \, dt \right)^{\frac{1}{2}}$$

$$\times \left( \int_0^{t_1} \int_{B_{2R}} |u - u_\delta^{ij}|^{\frac{2d+4}{d}} \, dx \, dt \right)^{\frac{d}{2d+4}}$$

$$\leq \tilde{C}_2 \delta^{\frac{1}{d+2}}.$$

Similarly, by the Hölder's inequality, Lemma 4 and Lemm 6 and Young's inequality, we find

$$I_3 \leq \left( \int_0^{t_1} \int_{B_{2R}} |\mathbf{b} - \mathbf{b}_\delta^i|^{\frac{d+4}{2}} \, dx \, dt \right)^{\frac{2}{2d+4}} \left( \int_0^{t_1} \int_{B_{2R}} |u_\delta|^{\frac{2d+4}{d}} \, dx \, dt \right)^{\frac{d}{2d+4}}$$

$$\times \left( \int_0^{t_1} \int_{B_{2R}} |Du - Du_\delta|^2 \, dx \, dt \right)^{\frac{1}{2}}$$

$$\leq \frac{\lambda}{16} \int_{B_{2R}} |Du - Du_\delta|^2 \, dx \, dt + \tilde{C}_2 \delta^{\frac{2}{d+2}}.$$

By the Hölder's inequality, we find

$$I_4 \leq \left( \int_0^{t_1} \int_{B_{2R}} |\mathbf{b} - \mathbf{b}_\delta^i|^{\frac{2d+4}{2}} \, dx \, dt \right)^{\frac{2}{2d+4}} \left( \int_0^{t_1} \int_{B_{2R}} |u_\delta|^2 \, dx \, dt \right)^{\frac{1}{2}}$$

$$\times \left( \int_0^{t_1} \int_{B_{2R}} |u - u_\delta|^{\frac{2d+4}{d}} \, dx \, dt \right)^{\frac{d}{2d+4}}$$

$$\leq \tilde{C}_2 \delta^{\frac{2}{d+2}}.$$

Following a similar procedure of estimating $I_3$, we find

$$I_5 \leq \frac{\lambda}{32} \int_{B_{2R}} |Du - Du_\delta|^2 \, dx \, dt$$

$$+ \tilde{C}_2 \delta^{\frac{2}{d+2}}.$$

Finally, we estimate

$$I_6 \leq \tilde{C}_2 \delta^{\frac{1}{d+2}}.$$

Using the fact $LHS = RHS$ and combining all the estimates, we find

$$\|u - u_\delta\|_{V_2(B_R)} \leq \frac{\tilde{C}_1}{R} + \tilde{C}_2 \delta^\alpha,$$

for some $\alpha = \alpha(d, \lambda, \Lambda) > 0$.

$\square$

## M  BARRON NORM ESTIMATE OF $u_\delta = K_\delta * f$ FOR ELLITPIC EQUATIONS

**Lemma 8.** *For some $\alpha > 0$, we denote*

$$\Psi_\alpha(x) = \frac{1}{(4\pi)^{\frac{d}{2}}} \int_0^\infty e^{-\alpha t} t^{-\frac{d}{2}} \exp(-\frac{|x|^2}{t}) \, dt.$$

*Then, we have*

$$\int_{\mathbb{R}^d} \Psi_\alpha(x) \, dx = \frac{1}{2^d \alpha},$$

$$\int_{\mathbb{R}^d} |x| \Psi_\alpha(x) \, dx = \frac{\sqrt{\pi} \Gamma(\frac{d+1}{2})}{2^{d+1} \alpha^{\frac{3}{2}} \Gamma(\frac{d}{2})}.$$

*Proof.* By substituting $t = |x|^2/s$, we have

$$\Psi_\alpha(x) = \frac{1}{(4\pi)^{\frac{d}{2}} |x|^{d-2}} \int_0^\infty s^{\frac{d}{2}-2} \exp(-\alpha \frac{|x|^2}{s} - s) \, ds.$$

We then use the integral change and the definition of the Gamma function to find

$$\int_{\mathbb{R}^d} \Psi_\alpha(x) \, dx = \int_0^\infty \int_{\partial B_r} \Psi_\alpha(x) \, dS dr$$

$$= \frac{2\pi^{\frac{d}{2}}}{\Gamma(\frac{d}{2})} \int_0^\infty \frac{r^{d-1}}{(4\pi)^{\frac{d}{2}} r^{d-2}} \int_0^\infty s^{\frac{d}{2}-2} \exp(-\alpha \frac{r^2}{s} - s) \, ds \, dr$$

$$= \frac{1}{2^{d-1}\Gamma(\frac{d}{2})} \int_0^\infty s^{\frac{d}{2}-2} e^{-s} \int_0^\infty r \exp(-\alpha \frac{r^2}{s}) \, dr \, ds$$

$$= \frac{1}{2^{d-1}\Gamma(\frac{d}{2})} \int_0^\infty s^{\frac{d}{2}-2} e^{-s} \int_0^\infty r \exp(-\alpha \frac{r^2}{s}) \, dr \, ds$$

$$= \frac{1}{2^d \Gamma(\frac{d}{2}) \alpha} \int_0^\infty s^{\frac{d}{2}-1} e^{-s} \, ds$$

$$= \frac{1}{2^d \alpha}.$$

Similarly, we have

$$\int_{\mathbb{R}^d} |x| \Psi_\alpha(x)\, dx = \int_0^\infty \int_{\partial B_r} r \Psi_\alpha(x)\, dS dr$$

$$= \frac{2\pi^{\frac{d}{2}}}{\Gamma(\frac{d}{2})} \int_0^\infty \frac{r^d}{(4\pi)^{\frac{d}{2}} r^{d-2}} \int_0^\infty s^{\frac{d}{2}-2} \exp(-\alpha \frac{r^2}{s} - s)\, ds\, dr$$

$$= \frac{1}{2^{d-1}\Gamma(\frac{d}{2})} \int_0^\infty s^{\frac{d}{2}-2} e^{-s} \int_0^\infty r^2 \exp(-\alpha \frac{r^2}{s})\, dr\, ds.$$

Using the fact that

$$\int_0^\infty t^2 exp(-\beta t^2)\, dt = \frac{\sqrt{\pi}}{4\beta^{\frac{3}{2}}},$$

we find

$$\frac{1}{2^{d-1}\Gamma(\frac{d}{2})} \int_0^\infty s^{\frac{d}{2}-2} e^{-s} \int_0^\infty r^2 \exp(-\alpha \frac{r^2}{s})\, dr\, ds.$$

$$= \frac{\sqrt{\pi}}{2^{d+1}\alpha^{\frac{3}{2}}\Gamma(\frac{d}{2})} \int_0^\infty s^{\frac{d-1}{2}} e^{-s}\, ds$$

$$= \frac{\sqrt{\pi}\Gamma(\frac{d+1}{2})}{2^{d+1}\alpha^{\frac{3}{2}}\Gamma(\frac{d}{2})}.$$

$\square$

BARRON NORM ESTIMATE OF THE FUNCTION $u_\delta = K_\delta * f$

*Proof.* Before we start, we can take $C_{scale} = 2^d$ from the result of Lemma 8. For a symmetric possitive definite matix $\mathbf{A}$, and coefficients of the lower order term $d(x)$, let us denote

$$\Psi_{d(x),\mathbf{A}}(x) = \frac{1}{(4\pi)^{\frac{d}{2}}|\det(\mathbf{A})|^{\frac{1}{2}}} \int_0^\infty e^{-d(x)t} t^{-\frac{d}{2}} \exp(-\frac{\langle \mathbf{A}^{-1}x, x \rangle}{t})\, dt.$$

As proceed in Appendix K, we decompose $\mathbf{A} = Q^T D^{\frac{1}{2}} D^{\frac{1}{2}} Q$. By Lemma 8 and substituting

$$z = D^{-\frac{1}{2}} Q^T x,$$

we find

$$\int_{\mathbb{R}^d} \Psi_{\alpha,\mathbf{A}}(x)\, dx = \int_{\mathbb{R}^d} \Psi_\alpha(z)\, dz = \frac{1}{2^d \alpha},$$

$$\int_{\mathbb{R}^d} |x| \Psi_{\alpha,\mathbf{A}}(x)\, dx = \int_{\mathbb{R}^d} |QD^{\frac{1}{2}}x| \Psi_\alpha(x)\, dx \le \frac{|\Lambda|^{\frac{1}{2}}\sqrt{\pi}\Gamma(\frac{d+1}{2})}{2^{d+1}\alpha^{\frac{3}{2}}\Gamma(\frac{d}{2})}.$$

Then $K_\delta(x, y)$ defined in equation 14 is

$$K_\delta(x, y) = C_{scale} \Psi_{d(x-y),\mathbf{A}_\delta}(x - y) \le C_{scale} \Psi_{d_{min},\mathbf{A}_\delta}(x - y).$$

If $u_\delta$ is the solution of equation 13, then it can be written as

$$u_\delta(x) = C_{scale} \Psi_{d(\cdot),\mathbf{A}_\delta} * f = C_{scale} \int_{\mathbb{R}^d} \Psi_{d(y),\mathbf{A}_\delta}(y) f(x - y)\, dy.$$

For simplicity, we set $f(x) = a\sigma(wrx + b)$. Then we have

$$u_\delta(x) = C_{scale} a \int_{\mathbb{R}^d} \sigma(w^\top(x - y) + b) \Psi_{d(y),\mathbf{A}_\delta}(y)\, dy.$$

Taking a Barron norm of $u_\delta$, we find

$$
\begin{aligned}
\|u_\delta\|_{\mathcal{B}(\mathbb{R}^d)} &\leq 2^d |a| \int_{\mathbb{R}^d} (|w| + |b|) \Psi_{d(y),\mathbf{A}_\delta}(y) + |w||y| \Psi_{d(y),\mathbf{A}_\delta}(y)\, dy \\
&\leq 2^d |a| \int_{\mathbb{R}^d} (|w| + |b|) \Psi_{d_{min},\mathbf{A}_\delta}(y) + |w||y| \Psi_{d_{min},\mathbf{A}_\delta}(y)\, dy \\
&\leq \frac{1}{d_{min}} (|a|(|w| + |b|)) + \frac{|\Lambda|^{\frac{1}{2}} \sqrt{\pi}\Gamma(\frac{d+1}{2})}{2 d_{min}^{\frac{3}{2}} \Gamma(\frac{d}{2})} |a||w| \\
&\leq \left( \frac{1}{d_{min}} + \frac{\Lambda^{\frac{1}{2}} \sqrt{\pi}\Gamma(\frac{d+1}{2})}{2 d_{min}^{\frac{3}{2}} \Gamma(\frac{d}{2})} \right) \|f\|_{\mathcal{B}(\mathbb{R}^d)}.
\end{aligned}
$$

$\square$

# N    COMPARISON WITH MODEL CASE

In this section, we compare our results with heat equation of the type. Consider the following

$$
\begin{cases}
u_t(t,x) - \Delta u(t,x) = f & \text{in} \quad (0,\infty) \times \mathbb{R}^d, \\
u(0,x) = g(x) & \text{in} \quad \mathbb{R}^d.
\end{cases}
\tag{26}
$$

Then (Weinan & Wojtowytsch, 2022, Lemma 3), we have

$$
\|u(t,\cdot)\|_{\mathcal{B}(\mathbb{R}^d)} \leq (1 + \sqrt{t})\|g\|_{\mathcal{B}(\mathbb{R}^d)} + \left( t + \frac{2}{3}t^{\frac{3}{2}} + \frac{t^2}{2} + \frac{5}{2}t^{\frac{5}{2}} \right) \|f\|_{\mathcal{B}((0,\infty) \times \mathbb{R}^d)}.
\tag{27}
$$

The result of Theorem 1 coincides with equation 27 after multiplying the left-hand side by 2 when $\mathbf{b} = \mathbf{c} = 0$ and $\Lambda = 1$. This can be viewed as a natural extension of the existing result in (Weinan & Wojtowytsch, 2022, Lemma 3).

Similarly, for the elliptic equation, Weinan & Wojtowytsch (2022) considers

$$
-\Delta u(t,x) + \lambda^2 = f \qquad \text{in} \quad \mathbb{R}^d.
\tag{28}
$$

By (Weinan & Wojtowytsch, 2022, Lemma 2), we have

$$
\|u\|_{\mathcal{B}(\mathbb{R}^d)} \leq (\lambda^{-2} + \lambda^{-3})\|f\|_{\mathcal{B}(\mathbb{R}^d)}.
\tag{29}
$$

The result of Theorem 2 coincides with equation 27 after multiplying the left-hand side by 2 when $d(x) = \lambda^2$ and $d = 3$. This can be viewed as a natural extension of the existing result in (Weinan & Wojtowytsch, 2022, Lemma 2).

# O    NUMERICAL EXPERIMENTS

In this section, we provide numerical experiments as examples to support our theoretical findings that a two-layer network can approximate the second-order solution without suffering from the curse of dimensionality. To this end, we consider the following equations:

$$
\begin{cases}
-\text{div}\,(a(x)\nabla u) = f & \text{in} \quad [0,1]^d, \\
u = 0 & \text{on} \quad \partial([0,1]^d)
\end{cases}
\tag{30}
$$

and

$$
\begin{cases}
u_t(t,x) - \text{div}\,(a(x)\nabla u(t,x)) = f & \text{in} \quad (0,1] \times [0,1]^d, \\
u(0,x) = g(x) & \text{in} \quad [0,1]^d, \\
u = 0 & \text{on} \quad \partial_p\left((0,1] \times [0,1]^d\right).
\end{cases}
\tag{31}
$$

Table 1: Values of $\|u\|_{\mathcal{B}([0,1]^d)}/\|f\|_{\mathcal{B}([0,1]^d)}$ across different dimensions and cases.

| Case | Dimension (d) | | | | |
|------|-----|------|-------|-------|--------|
| | d=2 | d=10 | d=100 | d=500 | d=1000 |
| Case 1 | 0.19 | 0.06 | 0.18 | 0.11 | 0.11 |
| Case 2 | 0.26 | 0.22 | 0.29 | 0.23 | 0.59 |

Table 2: Values of $\|u(1,\cdot)\|_{\mathcal{B}([0,1]^d)}/(\|f\|_{\mathcal{B}((0,1)\times[0,1]^d)} + \|g\|_{\mathcal{B}([0,1]^d)})$ across different dimensions and cases.

| Case | Dimension (d) | | | | |
|------|-----|------|-------|-------|--------|
| | d=2 | d=10 | d=100 | d=500 | d=1000 |
| Case 3 | 0.37 | 0.11 | 0.04 | 0.04 | 0.02 |
| Case 4 | 0.16 | 0.03 | 0.10 | 0.06 | 0.08 |

Let us present four cases where the closed-form solution, coefficient, and forcing term are known. To be precise, we consider the following four cases.

Case 1: Solution of equation 30 with

$$a(x) = 1, \quad u(x) = \frac{1}{\pi^2}\sum_{i=1}^{d}\sin(\pi x_i), \quad f(x) = \sum_{i=1}^{d}\sin(\pi x_i).$$

Case 2: Solution of equation 30 with

$$a_i(x) = 1 + \cos(\pi x_i), \quad u(x) = \frac{1}{\pi^2}\sum_{i=1}^{d}\sin(\pi x_i),$$

$$f(x) = \sum_{i=1}^{d}\sin(\pi x_i) + \sin(2\pi x_i).$$

Case 3: Solution of equation 31 with

$$a(x) = 1, \qquad\qquad\qquad u(t,x) = \frac{e^{-t}}{\pi^2}\sum_{i=1}^{d}\sin(\pi x_i),$$

$$f(t,x) = \frac{e^{-t}(\pi^2 - 1)}{\pi^2}\sum_{i=1}^{d}\sin(\pi x_i), \qquad g(x) = \frac{1}{\pi^2}\sum_{i=1}^{d}\sin(\pi x_i).$$

Case 4: Solution of equation 31 with

$$a_i(x) = 1 + \cos(\pi x_i), \qquad\qquad\qquad u(t,x) = \frac{e^{-t}}{\pi^2}\sum_{i=1}^{d}\sin(\pi x_i),$$

$$f(t,x) = \frac{e^{-t}(\pi^2 - 1)}{\pi^2}\sum_{i=1}^{d}\sin(\pi x_i) + e^{-t}\sum_{i=1}^{d}\sin(2\pi x_i), \qquad g(x) = \frac{1}{\pi^2}\sum_{i=1}^{d}\sin(\pi x_i).$$

Here, we denote $a_i$ as the $i$-th element of $a(x) = (a_1(x), \cdots, a_n(x)) \in \mathbb{R}^d$. For Case 1 and Case 2, we estimate $\|u\|_{\mathcal{B}([0,1]^d)}/\|f\|_{\mathcal{B}([0,1]^d)}$ for dimensions $2, 10, 100, 500$, and $1000$, as shown in Table 1. Similarly, we estimate $\|u(\cdot,1)\|_{\mathcal{B}([0,1]^d)}/(\|f\|_{\mathcal{B}((0,1)\times[0,1]^d)} + \|g\|_{\mathcal{B}([0,1]^d)})$ in Table 2 for dimensions $2, 10, 100, 500$, and $1000$. To numerically calculate the Barron norm, we approximate each function with a two-layer network and compute the path-norm defined in equation 7.

# P    ESTIMATES ON $\Gamma(\frac{d+1}{2})/\Gamma(\frac{d}{2})$

By the (Gubner, 2021, equation 3), we have

$$Cx^{x-\frac{1}{2}}e^{-x} \le \Gamma(x) \le Cx^{x-\frac{1}{2}}e^{-x}e^{\frac{1}{12x}}$$

for some constant $C > 0$. Then we have

$$\frac{\Gamma(\frac{d+1}{2})}{\Gamma(\frac{d}{2})} \le \frac{C(\frac{d+1}{2})^{\frac{d}{2}}e^{-\frac{d+1}{2}}e^{\frac{1}{6(d+1)}}}{C(\frac{d}{2})^{\frac{d-1}{2}}e^{-\frac{d}{2}}}$$

$$= e^{-\frac{1}{2}}e^{\frac{1}{6(d+1)}}\sqrt{\left(1+\frac{1}{d}\right)^d}\sqrt{\frac{d}{2}} \approx \sqrt{\frac{d}{2}}$$

for large $d \in \mathbb{N}$.

