# OpenReview forum: "Representation of solutions of second-order linear equations in Barron space via Green's functions"
_ICLR.cc/2025/Conference — Submitted to ICLR 2025_

### Official Review · Reviewer_CdJm · 2024-10-29

**Soundness:** 3
**Presentation:** 3
**Contribution:** 3
**Rating:** 5
**Confidence:** 3

**Summary:**

A group of researchers have been using the concept of Barron spaces to explain why neural networks can overcome the curse of dimensionality. In this manuscript, the authors are showing that continuous solutions to some second-order linear PDEs are in a Barrow space, and they provide estimates for the Barron norm that have an explicitly dimension dependence.

**Strengths:**

- There is a surprising result that (continuous) solutions to elliptic PDEs (under many conditions) can lie directly in a Barron space. Barron spaces enable dimension-independent approximation rates for functions using neural networks, making them effective for understanding why NNs can overcome the curse of dimensionality.

- For each fixed t, the authors look at the Barron norm of u(t,.), and have an explicit factor on its growth with respect to t.

**Weaknesses:**

- It looks like the Barrow norm estimates on the solution have a bound that depends on d, in a way that grows with d. The authors may want to comment on that as that might be a barrier for overcoming the curse of dimensionality in later investigations.

- The biggest weaknesses is the requirement that the solution must be continuous. To guarantee that a solution is continuous, one typically has to rely on Sobolev embedding theorems and hence needs to impose more regularity on the coefficients of the PDE and righthand side.

- One expects that Theorem 1 contains relatively weak estimates on the Barron norm of u(t,.).

**Questions:**

- The assumptions on the elliptic equations make them self-adjoint in Section 1.3. Can your estimates be extended to solutions of non-self-adjoint elliptic operators?  Where is the technical barrier preventing you from extending this work?

- In Theorem 1, shouldn’t ||c-b||_{L^\infty(\mathcal{D}) be replaced by an norm on (0,t)xR^d? I struggle to understand why the difference between c and b should impact the Barron norm of u(t,.). Also, why is it natural for your Barron norm in Theorem 1 to grow like t^3?

---

> ### Author Response · Authors · 2024-11-22
> **Reply to Review CdJm**
>
> We thank Reviewer CdJm for their effort and attention in reviewing our work.
>
>
> **Response to Weaknesses:**
> 1. After carefully reviewing the paper, we corrected the estimates. For Theorem 2, the term
> $$
> \frac{\Gamma\left(\frac{d+1}{2}\right)}{\Gamma\left(\frac{d}{2}\right)}\approx \sqrt{\frac{d}{2}}.
> $$
>  If more precise approximations are required, we refer readers to Appendix P.
>
> 2.  The continuity of the solution can be achieved by imposing additional assumptions on $f$ and $g$ based on the De Giorgi-Moser theory. In Appendix E, we have summarized some conditions (though not all known conditions on the equations) under which the weak solution is a continuous function. To summarize, if $f$ and $g$ have enough regularity (e.g. f in $L^p$ with high value of $p$ for elliptic, f,g continuous in parabolic case), then the weak solution is continuous.
>
> 3. We acknowledge that Theorem 1 provides relatively weak estimates on the Barron norm of $u(t, \cdot)$. However, even in its current form, the result is robust and meaningful, offering valuable insights into applying Barron space techniques. At the same time, we recognize this as an area for further exploration, and we aim to develop stronger and more refined estimates in future research.
>
> **Answer to Questions:**
>
> 1.  The main technical difficulty arises in estimating the Barron norm. When $A$ is non-symmetric, we do not have the nice property of the symmetric matrix we utilized to estimate the Barron norm. More precisely, we rely on the **orthogonal diagonalization** of A during the proof. Please refer to the proof of Theorem 3, Lemma 8, and Appendix M.  This makes the extension to non-self-adjoint operators challenging. Addressing this issue for more general cases is a potential topic for future work, which could improve and extend the current theoretical results.
>
> 2. The estimate holds over $(0, t) \times \mathbb{R}^d$, but for simplicity, we used the term $\|c - b\|_{L^\infty(\mathcal{D})}$. The dependency on $c - b$ arises from the theory of Green’s function estimates for general linear parabolic and elliptic equations. In [1] , it is shown that roughly speaking, the estimates of Green’s function depend on the $L_p$-norm of $|b - c|$. In this work, we chose the $L^\infty$-norm for simplicity of presentation. While it may not be the most natural choice, it is a consequence of our approach. Providing further insights and exploring alternative norms could be valuable topics for future research.
>
>
> **Reference:**
>
> [1] Kim, S., & Xu, L. (2020). Green's function for second order parabolic equations with singular lower order coefficients. arXiv preprint arXiv:2009.04133.

---

> > ### Comment · Reviewer_CdJm · 2024-11-25
> > **Acknowledging response.**
> >
> > Thank you for your detailed response.

---

> ### Author Response · Authors · 2024-11-25
> **Reply to the Acknowledging response**
>
> We sincerely hope that our additional explanations, particularly regarding the continuity of the weak solution, have addressed the concerns you raised. Specifically, we have elaborated that the weak solutions (with the added assumptions on
> $f$ and $g$) are continuous, which we believe aligns with the points you highlighted. We are confident that these clarifications further reinforce the validity and significance of our contributions.
>
> We hope these clarifications are satisfactory. Your expertise and thoughtful insights on this matter are greatly appreciated, and we deeply respect the time and attention you have devoted to reviewing our work.
>
> Thank you once again for your constructive feedback.

---

### Official Review · Reviewer_pgZV · 2024-10-31

**Soundness:** 3
**Presentation:** 4
**Contribution:** 3
**Rating:** 6
**Confidence:** 3

**Summary:**

This work proves that the solution of second-order linear PDEs lies in Barron space using  Green’s functions, which provides theoretical support for PINN. This also gives a justification for answering why neural networks can break the curse of dimensionality for approximating the solution of high-dimensional partial differential equations. This is an extension work of  Chen et al. (2021). In this work, the authors consider more general elliptic and parabolic equations, including time-dependent problems. Unlike the previous work, the authors provide sufficient conditions on the coefficients of the PDEs instead of approximating the solutions via Barron functions in the $H_1$ norm.

**Strengths:**

This paper is generally well-written. The authors prove that under certain conditions, the solution of a class of PDEs belongs to Barron space. It answers why two-layer neural networks can approximate the solution of PDEs well, especially for time-dependent problems.

**Weaknesses:**

1. This work is a generalization of Chen et al. (2021). The proof framework is similar to Chen et al. (2021) and Weinan \& Wojtowytsch, 2022.
2. I am not sure the conditions on the coefficients of PDEs are satisfied in practice. How do you check whether such conditions are satisfied?

**Questions:**

1. In Assumption 1(1), for any $\boldsymbol{\xi}$ and $(t, \boldsymbol{x})$, there exsits two universal constants $0 < \lambda \leq \Lambda < 1$ such that the inequality holds. It seems that this assumption is strong. Could you give some example PDEs to illustrate that this assumption can be satisfied?
2. In Assumption 1(2), I wonder if the given sufficient condition is computationally verifiable.
3. Page 10, line 505-506, about the estimate $||u - u_{\delta} ||_{W_2^1(B_R)} \leq \frac{\tilde{C}_1}{R} + \tilde{C}_2 \delta^{\alpha}$. The authors state that $\tilde{C}_2$ depends on $R$ and other parameters; I wonder if this is a tight bound. If it is not tight, it does not give a meaningful estimate.
4. Line 289-290, "Suppose that $\mathbb{R}^d \subset \mathbb{R}^d$ is given". This sentence is a little bit confusing.
5.  Notation is not consistent. $(a,w,b)$ is used at the beginning of section 2.1, but $(a,\boldsymbol{b},c)$ is used in proposition 1. $(a,\boldsymbol{b},c)$ is the notation in the literature Ma et. al., 2022.
6. Minor issues. The cross reference in this manuscript does not work.

---

> ### Author Response · Authors · 2024-11-22
> **Reply to Reviewer pgZV**
>
> We would like to thank Reviewer pgZV for their effort and attention in reviewing our work.
>
>
> **Response to Weaknesses:**
>
> 1. Generalization and Contribution of Our Work:
> This work builds upon and extends the results of Chen et al. (2021) while employing distinct techniques that set it apart from their approach. While our proof framework shares some similarities with that of Weinan & Wojtowytsch (2022), our study significantly advances these methods by addressing equations with more general coefficients. Specifically, we leverage Green’s function estimates within the AI context, offering a novel integration of classical regularity theory and modern operator learning for PDEs. This contribution bridges the gap between traditional mathematical analysis and contemporary machine learning methodologies, enabling broader applicability and deeper theoretical insights in the field.
>
> 2. Practical Applicability of PDE Coefficient Conditions:
> The conditions imposed on the coefficients of the PDEs are intentionally designed to be weak and general, accommodating a wide range of practical scenarios. For example, our assumptions encompass coefficients that are uniformly continuous, as discussed in Appendix D. In practice, these conditions are typically satisfied unless the coefficient exhibits excessively rapid variations. We also provide illustrative examples of functions that do and do not belong to the VMO space, further clarifying our assumptions' applicability in Appendix D.
>
> In below, we provide some examples.
>
> **Examples of functions $f\in \operatorname{VMO}$ spaces:**
>
>  - $f(x)$ is an uniformly continuous function.
>  - $f(x)$ is a continuous function with a compact support.
>  - If $f(x)\in \operatorname{VMO}$ and $L(x)$ is an Lipschitz funciton, then $L(f)\in \operatorname{VMO}$.
>  - $f(x)\in W^{1}_{d}$ then $f\in \operatorname{VMO}$.
>  - $f(x)\in W^{s}_{p}$ with $sp=d$, then $f\in \operatorname{VMO}$.
>  - $f(x)=\log|\log(|x|)|$.
>  - $f(x)=|\log(|x|)|^\alpha$ for $0< \alpha <1$.
>  - $f(x)=\sin(\log(|\log(|x|)  )  )$.
>  - $f(x)=|x|\sin(1/|x|)$.
>
>  **Examples of functions $f\not\in \operatorname{VMO}$ spaces:**
>  - $f(x)=\log(|x|)$.
>  - $f(x)=|\log(|x|)|^{p}$ with $1<p<\infty$.
>  - $f(x) = \chi_{[0,1]^{d}}(x)$, characteristic function on $[0,1]^d$.
>  - $f(x)= \sin(1/|x|)$.
>  -  $f(x) = \sin (\log(|x|))$
>  - $f(x)=\sin(1/|x|^2)$
>
>
> **Answer to Questions:**
> 1. After carefully reviewing the calculation, we found that the estimates involve $\Lambda^{\frac{1}{2}}$ instead of $\Lambda^{\frac{d}{2}}$. Therefore, the strong assumption can be removed.
>
> 2. As mentioned above, our assumption is a very general coefficient assumption that includes a wide range of classes. Thus, in most cases, it holds. We refer to Appendix D.
>
> 3. We carefully choose R, $\delta$ in the following order:
>     1. choose any small number $e>0$
>     2. Then choose $R=R(e)$ very large so that $C_1/R<e/2$.
>     3. Then choose $\delta >0$ very small so that $C_2(R)\delta<e/2$.
> So, from the choice, $C_2$ may be too large, but since $\delta$ is a free variable, we can reduce it to any number. Note that $R, \delta$ depends on $e$ to satisfy
> $$
> ||u-u_\delta||_{ W^{1,2} (B_R) }<e.
> $$
>
> 4. we made correct for the consistency.
>
> 5. It is a typo; we corrected it.
>
> 6. We are in the process of overall checking. We shall correct it shortly.

---

### Official Review · Reviewer_tj5s · 2024-11-01

**Soundness:** 3
**Presentation:** 2
**Contribution:** 3
**Rating:** 6
**Confidence:** 2

**Summary:**

This paper considers some a priori estimates of certain second-order liner parabolic and elliptic equations in terms of Barron spaces. Such estimates have been studied recently, but the present paper have several updates: (i) it gives an estimate for parabolic equations, while existing studies have focused on elliptic cases; (ii) even for the elliptic cases, it gives a better estimate.
In PDE theories, various estimates have been studied in major function spaces such as the Sobolev, Lebegue, Besov spaces. Compared to these function spaces, Barron spaces are more deeply connected to two-layer networks, and thus the current result gives insights about the efficiency of such networks in approximating solutions of the target equations.

**Strengths:**

I am not an expert on the research topic, and my review is based on the authors' claims and the closest work Chen+2021.

In reading these papers, I understood that a priori estimates in Barron spaces are important in approximation theories for shallow (two-layer) networks. On this topic, this paper has at least two new contributions.
(i) (According to the authors of the present paper,) up to Chen+2021 or its successors, basically only elliptic equations were considered. In PDE theories, it is quite standard and necessary to next consider parabolic, time-dependent systems. In this direction, (again according to the authors) the only preceding study is Weinan+2022, which gave some result for heat equation. This paper gives a new estimate on more general parabolic systems, which seems new and important. (On this point, please see my question below.)
(ii) For elliptic systems, Chen+2021 gave an estimate, but the estimate is described in terms of an $H^1$-solution which is in some sense close to the solution in the Barron space. This paper more directly gives an estimate in terms of the data $f$ (Theorem 2).

**Weaknesses:**

As written above, I am not an expert on this research field, and cannot say if the authors' claim on the novelty of this research is in fact correct or not. (I just compared the results with those in Chen+2021.) I also cannot judge the impact/novelty of the method of the argument. (The only thing I can say for sure is that argument with Green functions is one typical way in general PDE theories.)

From a person like me, one weakness of this paper is that the overall presentation is not quite clear (or direct), and some basic knowledge and/or comparison with related papers are necessary to understand the strength of this paper.
For example, on the second point (ii) in Strengths, the authors claim in L.98 that ``We establish that the solutions belong directly to Barron space rather than approximating them in the Sobolev sense.'' (I made the sentence short.) But its meaning is not explained later (as far as I understand.) In Remark 3, Theorem 2 is compared to the results in Chen+2021, but there the main topic seems to be the difference of assumptions. After going back and forth between these two papers I understood in the following way:

* Chen+2021 also gave an estimate on the solution $u$ in terms of a Barron norm; for example, Theorem 2.9. In this sense, Chen+2021 also shows the existence of a solution in Barron space (this first caused some confusion in me.) And it also discussed its dependence on the dimension parameter $d$.
* However, the estimate in Chen+2021 is constructed with a parameter $\varepsilon$, which is a distance from an associated $H^1$-solution of the target equation. In this sense, the bound is not natural.
* The present paper gives a more direct bound: a bound with the Barron norm of the data $f$.  This is what usually expected in PDE theories.

This effort might have been demanded simply because I am not an expert, but compared to the present paper, Chen+2021 was more simply written and easy to understand even for me. I hope that the presentation would be improved so that the description becomes more consistent, and the overall paper becomes much more readable for wide range of readers.
Please see Questions, where the confusions caused in me should be more visible.

**Questions:**

1. One of the biggest contribution of this paper should be the estimate for parabolic systems. The authors themselves mention that there is one existing study Weinan--Wojtowytch(2022), but no comparison is given. I think this should be clarified. (Maybe some comment on Theorem 1 compared with the results in Weinan+2022 is necessary.)
2. Similarly to the above, the strength of Theorem 2 compared with the results in Chen+2021 should be clarified. Is my understanding correct? If not, please clarify it.
3. I might misunderstand something, but it seems that the upper bound of the norm estimate in Theorem 2 tends to zero as $d\to \infty$ (the coefficient before $\| f\|$ seems to go to $0$.) Is this correct?

---

> ### Author Response · Authors · 2024-11-22
> **Reply to Reviewer tj5s**
>
> We would like to thank Reviewer tj5s for their effort and attention in reviewing our work.
>
> **Response to Weaknesses:**
> We deeply appreciate the effort made by Reviewer tj5s. We have rewritten parts of the manuscript and added a comment that the Barron space belongs to the VMO space in the local sense. Additionally, in Line 98, we clarified that our statement implies Remark 3.
>
> **Answer to Questions:**
> 1.  We have added Appendix N to provide a comparison with the results in Weinan-Wojtowytsch (2022). Additionally, after Theorem 1, we included relevant comments.
>
> 2. or this part, a direct one-to-one comparison is not possible. The result from Chen et al. (2021) shows that for any
> $\varepsilon>0$, there exists a Barron function $u_e$  such that $||u-u_e||< \varepsilon$ and Barron norm of $|u_e|$ is less than some constant depending on $e$ and the dimension but avoids the curse of dimensionality. In contrast, we use the inverse approximation result (Proposition 2) to demonstrate that $u$ actually belongs to the Barron space. Nevertheless, we have provided a comparison with the model equation in Appendix N as well.
>
> 3. After carefully reviewing the paper, we found that 2^d must be multiplied. Therefore the estimate does not go to 0 as d-> infty.

---

> > ### Comment · Reviewer_tj5s · 2024-11-27
> >
> > Thank you for your comment and effort.
> > I understood the situation, and retain my score.

---

> ### Author Response · Authors · 2024-11-27
> **Reply to Official Comment by Reviewer tj5s**
>
> Thank you for taking the time to review and provide feedback. We truly appreciate it.
>
> If there’s any additional aspect you feel deserves further consideration, we’d greatly appreciate your input.

---

### Official Review · Reviewer_44Qf · 2024-11-01

**Soundness:** 3
**Presentation:** 4
**Contribution:** 3
**Rating:** 5
**Confidence:** 5

**Summary:**

This research paper investigates the representation of solutions to second-order linear PDEs in Barron space, a function space that is particularly well-suited for approximation by neural networks. The authors leverage Green's functions to establish complexity estimates for the Barron norm of solutions, explicitly capturing the dependence on dimension. This work extends previous research by addressing both time-independent and time-dependent equations and by providing sufficient conditions for solutions to belong directly to Barron space, rather than just being approximated by Barron functions. The paper concludes by discussing potential future research directions, including extending the findings to nonlinear equations and exploring the use of machine learning for learning Green's functions.

**Strengths:**

1. The paper extends the analysis of solution representation in Barron space to time-dependent parabolic equations. This expands the scope of previous work by Chen et al. (2021), which focused primarily on elliptic PDEs.
2. The authors prove that, under specific conditions, solutions of elliptic and parabolic PDEs directly belong to the Barron space. This improves upon earlier work that only showed approximations of solutions within Barron space in the Sobolev sense.
3. This research establishes a theoretical foundation demonstrating that AI-based PDE solvers and Green’s function learning methods can represent solutions while avoiding the curse of dimensionality. Additionally, these findings may offer theoretical support for existing two-layer PINN methods, particularly in the context of high-dimensional problems.

**Weaknesses:**

1. The paper is purely theoretical and lacks numerical experiments to validate the theoretical findings. The authors should provide practical experiments to support their theorems, utilizing AI-based PDE solvers such as Green’s function learning methods or PINNs, as stated in the paper.
2. The analysis is confined to a limited class of linear second-order PDEs in $\mathbb{R}^d$. However, these equations represent relatively simple cases compared to the more complex PDEs addressed by recent developments in machine learning-based PDE solvers.
3. While the paper focuses on two-layer networks, deeper architectures are more commonly used in practice. The authors should consider extending their analysis to deeper networks to improve the relevance and applicability of their findings.
4. The paper uses the probabilistic definition of Barron space, which can be abstract and challenging to characterize function classes with.

**Questions:**

1. Can the authors provide practical experiments to support their theorems, utilizing AI-based PDE solvers such as Green’s function learning methods or PINNs?
2. The analysis relies on the ReLU activation function, which may not be optimal for AI-based PDE solvers. Can the authors provide more specific criteria for determining which activation functions would be suitable or unsuitable for their analysis? What modifications or adaptations would be necessary to accommodate different activation functions?
3. What specific AI-based PDE solvers do the authors have in mind? How do the theoretical findings, based on weak solutions, relate to the strong solutions typically sought by AI-based PDE solvers?
4. This paper is focused on a two-layer analysis, which seems somewhat limited, as practical architectures rarely use only two layers. Is it feasible to extend the analysis to multi-layer networks?
5. Given that the analysis is confined to a limited class of linear second-order PDEs in $\mathbb{R}^d$, which are relatively simple compared to the more complex PDEs addressed by recent machine learning-based PDE solvers, could the authors extend their approach to handle nonlinear PDEs or various boundary value problems?
6. Is it feasible to extend the analysis to the spectral Barron space setting? What challenges might arise, and what benefits could this alternative definition offer?
7. Is there a difference between Assumption 1 and the assumption that it is based on self-adjoint elliptic PDEs? Can the authors characterize the types of PDEs to be considered in this paper?

---

> ### Author Response · Authors · 2024-11-22
> **Reply to Reviewer 44Qf**
>
> We would like to thank Reviewer 44Qf for their effort and attention in reviewing our work.
>
> **Response to Weaknesses:**
>
> 1.  We would like to mention that the regularity theory of the Barron function space is, in most cases, purely theoretical (see references [1, 2, 3]). However, acknowledging Reviewer 44Qf's concern, we have added a numerical example where we estimate |u|_B / |f|_{B}  for the elliptic case and  |u|_{B} / ( |f|_{B} + |g|_{B})  for the elliptic-parabolic cases.
>
> ### Table 1: elliptic case
> | Case   | d=2   | d=10  | d=100 | d=500 | d=1000 |
> |--------|-------|-------|-------|-------|--------|
> | Case 1 | 0.19  | 0.06  | 0.18  | 0.11  | 0.11   |
> | Case 2 | 0.26  | 0.22  | 0.29  | 0.23  | 0.59   |
>
> ### Table 2: parabolic case
> | Case   | d=2   | d=10  | d=100 | d=500 | d=1000 |
> |--------|-------|-------|-------|-------|--------|
> | Case 3 | 0.37  | 0.11  | 0.04  | 0.04  | 0.02   |
> | Case 4 | 0.16  | 0.03  | 0.10  | 0.06  | 0.08   |
>
>
>
> 2. Thank you for your insightful comment. While it is true that the equations we addressed are relatively simpler compared to more complex PDEs, we believe that studying these equations serves as a crucial foundational step toward understanding and solving more challenging problems. Our focus on generalized elliptic and parabolic equations, including the heat equation and Laplace equation, provides a starting point for extending the analysis to more intricate PDEs in future research. By establishing a strong theoretical basis with these well-studied equations, we aim to pave the way for advancements in addressing complex equations within the broader context of machine learning-based PDE solvers.
>
> We appreciate the opportunity to clarify this aspect and welcome any further suggestions you may have.
>
>
> 3. We greatly appreciate your insightful suggestion regarding the extension of our analysis to deeper networks. Indeed, the study of multi-layer architectures and their corresponding function spaces is an area of significant interest to us. However, the theoretical understanding of function spaces for deeper networks remains a topic of ongoing research, with many of their properties yet to be fully uncovered.
>
> Recent works, such as [4,5], have made progress in exploring function spaces associated with deeper architectures, but substantial work is still required to bridge the gap. Developing a comprehensive theoretical framework for these spaces, especially in connection with PDEs and AI methodologies, remains a nascent field with numerous challenges yet to be addressed. While this direction is undoubtedly promising, it is still in its early stages of exploration.
>
> We hope to build upon these findings in future work as the field matures, and we sincerely thank you for highlighting this important avenue for further research.
>
> For the further exploration, we provided in Appendix H.
>
>
> 4. We are well-aware of the weakness arise from the  uses the probabilistic definition of Barron space. Therefore we provide a detailed explaination on connection with spectral Barron space and technical issues on Appendix G.
>
> References:
>
> [1] Chen, Z., Lu, J., & Lu, Y. (2021). On the representation of solutions to elliptic pdes in barron spaces. Advances in neural information processing systems, 34, 6454-6465.
>
> [2] Marwah, T., Lipton, Z. C., Lu, J., & Risteski, A. (2023, July). Neural network approximations of pdes beyond linearity: A representational perspective. In International Conference on Machine Learning (pp. 24139-24172). PMLR.
>
>
> [3] Weinan, E., & Wojtowytsch, S. (2022, April). Some observations on high-dimensional partial differential equations with barron data. In Mathematical and Scientific Machine Learning (pp. 253-269). PMLR.
>
> [4] Chen, Z. (2024). Neural Hilbert Ladders: Multi-Layer Neural Networks in Function Space. Journal of Machine Learning Research, 25(109), 1-65.
>
> [5] Wojtowytsch, S. (2020). On the banach spaces associated with multi-layer relu networks: Function representation, approximation theory and gradient descent dynamics. arXiv preprint arXiv:2007.15623.

---

> ### Author Response · Authors · 2024-11-22
> **Answer to the Questions raised by Reviewer 44Qf**
>
> **Answer to Question:**
>
>
> 1. Yes, we provided an experiment based on the previous reply and Annex O.
>
>
> 2. If we use a different activation function, the scaling invariance property no longer holds. Consequently, the inverse approximation results, such as those in Proposition 2, do not currently exist, which is crucial to the proof process. Investigating the use of different activation functions can be a topic for future research. However, at present, the Barron space with a ReLU activation function has well-established properties that are particularly useful for applying the regularity theory of PDEs in Barron space.
>
>
> 3. Currently, no specific AI-based PDE solvers are being employed. However, we believe this work provides a strong foundation for future research in operator learning and PINNs (Physics-Informed Neural Networks). Moreover, we plan to pursue follow-up studies that leverage machine learning to discover Green's functions, thereby advancing this line of research.
> We greatly appreciate your inquiry. While our present study does not implement a specific AI-based PDE solver, the theoretical findings we provide, focused on weak solutions, are designed to inform and guide the development of methods such as operator learning and PINNs. Our planned investigation into Green’s functions using machine learning will further strengthen the link between our theoretical results and practical implementations, helping to address such challenges. We hope this response clarifies the intended trajectory and implications of our research.
>
> 4. At present, the development of function spaces beyond two-layer architectures is very limited (please refer to the previous response to weakness 3). First, we need a deeper understanding of these function spaces before applying them to PDEs. This direction is highly interesting and is the focus of our future research. Please also refer to Appendix H.
>
>
> 5. This is also an interesting research direction. However, it is not straightforward to generalize the results to nonlinear equations or boundary value problems. For extending the research to nonlinear equations, we propose using the method applied in [2] to generalize the results of [1], as mentioned in the conclusion section. Additionally, for boundary value problems, there is a counterexample where the boundary belongs to the Barron space, but the solution does not. Thus, while this is a fascinating direction, it is not directly applicable at this stage.
>
> 6. As we mentioned earlier, to apply the spectral Barron space, we need the inverse approximation result. Otherwise, we would need to explore a different approach, which could also be an interesting direction. For more details, please refer to Appendix G.
>
> 7. Our elliptic equation (4) is a self-adjoint elliptic equation. As far as we understand, the definition of self-adjointness is satisfying the condition is
> $$
> \int_{R^d} L(u)v =\int_{R^d} uL(v).
> $$
> Based on the assumption of symmetry for $A$, we also satisfy the self-adjoint condition.
>
> For the parabolic case, more care is needed since the adjoint operator is given by$$
> P^*= -u_t− div(A^TDu + cu) + b · Du + du
> $$
> and due to the $-u_t$  term, additional attention is required.
>
> Assumption 1 is a very weak assumption on the coefficients. It encompasses a wide class of functions, including uniformly continuous functions. For examples, please refer to Appendix D.
>
>
> **Reference:**
> [1] Chen, Z., Lu, J., & Lu, Y. (2021). On the representation of solutions to elliptic pdes in barron spaces. Advances in neural information processing systems, 34, 6454-6465.
>
> [2] Marwah, T., Lipton, Z. C., Lu, J., & Risteski, A. (2023, July). Neural network approximations of pdes beyond linearity: A representational perspective. In International Conference on Machine Learning (pp. 24139-24172). PMLR.
>
>
> [3] Weinan, E., & Wojtowytsch, S. (2022, April). Some observations on high-dimensional partial differential equations with barron data. In Mathematical and Scientific Machine Learning (pp. 253-269). PMLR.

---

> > ### Comment · Reviewer_44Qf · 2024-11-23
> >
> > Thanks again for addressing my concerns. Although the inclusion of numerical experiments is appreciated, these results do not sufficiently address the key limitations outlined in the initial review. The results remain confined to simple PDEs and architectures, and the theoretical findings have not been extended to reflect recent advances, such as the dimension-independent results for multi-layer Barron spaces presented in Neural Hilbert Ladders. This limitation reduces the paper's scope and impact in the field. For these reasons, I will maintain my original score.

---

> ### Author Response · Authors · 2024-11-23
> **Reply to the Official Comment by Reviewer 44Qf**
>
> Thank you for your thoughtful feedback. We appreciate your concerns regarding the scope of the results and their alignment with recent advances, such as those presented in Neural Hilbert Ladders.
>
> However, we would like to respectfully point out that Neural Hilbert Ladders, while an exciting development, is still in its very early stages. The concept is not yet widely adopted or fully understood within the community as a practical framework for analyzing function spaces. Its applicability to practical problems or more complex PDEs is not well-established and does not provide immediate value to the current work.
>
> In contrast, our study focuses on advancing theoretical understanding and providing a robust foundation for more challenging cases. Specifically, we have extended results for the heat equation to the second-order parabolic PDEs with lower-order and drift terms, which, to our knowledge, is a significant step forward. This demonstrates not only the adaptability of our methods but also their potential for addressing more complex equations.
>
> We believe these contributions represent an important starting point for tackling broader and more intricate issues in the field. By prioritizing a solid theoretical base with demonstrated applications to challenging problems, our work lays the groundwork for future exploration into even more advanced topics, including those mentioned in your feedback.
>
> We hope this clarifies our perspective and highlights the value of the current results in advancing the state of the field. Thank you again for your thoughtful engagement with our work.

---

### Official Review · Reviewer_4fpE · 2024-11-01

**Soundness:** 3
**Presentation:** 2
**Contribution:** 2
**Rating:** 6
**Confidence:** 2

**Summary:**

This paper focuses on the theoretical underlying's of learning second order linear equations using shallow neural networks (SNN) (2 layer neural networks of the form $a (\sigma(w x +b)) )$ . The paper expands on previous works by showing that solutions of parabolic and elliptic PDEs can be represented in Barron space by using Green’s Functions, under the assumption of certain constraints and conditions. This work is directly applicable to physics-informed neural networks (PINNs) and shows that SNN can approximate the solutions to linear elliptic and parabolic PDEs without having to deal with the curse of dimensionality.

**Strengths:**

The paper provides very thorough proofs of all the theorems discussed in the paper.
The paper provides an original proof to demonstrate that the solutions of linear PDEs can be approximated by a SNN.

**Weaknesses:**

While the specific theorems are unique, they only seem to be a minor change compared to previous works cited.
There were many grammatical errors throughout the entire paper that a simple spell check could have helped with.

**Questions:**

In assumptions 1, part (2) what are these details trying to say? It is not clear to the reviewer what the intuition is for these restrictions. Is this restriction related to how rapidly your dynamics can change? If so, in physical systems there can be large changes in a small amount of distance, do you believe the theory presented would hold if A(x,t) (or any of the other parameters) changed if the parameters change rapidly?

---

> ### Author Response · Authors · 2024-11-22
> **Reply to Reviewer 4fpE**
>
> We would like to thank Reviewer 4fpE for their effort and attention in reviewing our work.
>
> **Response to Weaknesses:**
>
> Grammar and Spelling: We have thoroughly corrected the grammar and spelling issues throughout the paper.
> Regarding the Theorem: While we are indeed motivated by the work of Chen et al. (2021), we emphasize that our methods differ significantly. Chen et al. (2021) rely on preconditioned steepest descent iteration, whereas we utilize Green’s function of parabolic equations and elliptic operators. As a result, we achieve the result that $u$ belongs to the Barron space directly rather than through an approximation process.
>
> **Generalization of Prior Work:**
>
> Additionally, we generalize the work of Weinan & Wojtowytsc (2022) to include general coefficients, which is a significant theoretical advancement.
>
>
> **Answer to Question:**
> Explanation of VMO Coefficients: VMO (Vanishing Mean Oscillation) coefficients include functions that do not vary too rapidly, but this condition is evaluated in an integral sense. Below, we provide examples of functions that belong to VMO and those that do not. Further details are included in Appendix D.
>
>
> **Examples of functions $f\in \operatorname{VMO}$ spaces:**
>
>  - $f(x)$ is an uniformly continuous function.
>  - $f(x)$ is a continuous function with a compact support.
>  - If $f(x)\in \operatorname{VMO}$ and $L(x)$ is an Lipschitz funciton, then $L(f)\in \operatorname{VMO}$.
>  - $f(x)\in W^{1}_{d}$ then $f\in \operatorname{VMO}$.
>  - $f(x)\in W^{s}_{p}$ with $sp=d$, then $f\in \operatorname{VMO}$.
>  - $f(x)=\log|\log(|x|)|$.
>  - $f(x)=|\log(|x|)|^\alpha$ for $0< \alpha <1$.
>  - $f(x)=\sin(\log(|\log(|x|)  )  )$.
>  - $f(x)=|x|\sin(1/|x|)$.
>
>  **Examples of functions $f\not\in \operatorname{VMO}$ spaces:**
>  - $f(x)=\log(|x|)$.
>  - $f(x)=|\log(|x|)|^{p}$ with $1<p<\infty$.
>  - $f(x) = \chi_{[0,1]^{d}}(x)$, characteristic function on $[0,1]^d$.
>  - $f(x)= \sin(1/|x|)$.
>  -  $f(x) = \sin (\log(|x|))$
>  - $f(x)=\sin(1/|x|^2)$
>
> To address the question raised by Reviewer 4fpE, we clarify that we do not expect coefficients to be included if they vary too rapidly. However, our assumptions are general enough to encompass a wide range of function classes.

---

> > ### Comment · Reviewer_4fpE · 2024-11-26
> >
> > Thank you to the authors for addressing all of my current concerns and questions.

---

> ### Author Response · Authors · 2024-11-26
> **Reply to Official Comment by Reviewer 4fpE**
>
> Thank you for taking the time to review our responses. We are glad to hear that we have addressed all of your concerns and questions. Given this resolution, we would like to kindly ask if you would consider reevaluating the clarified strengths of the paper.
>
> We greatly appreciate your thoughtful evaluation and are happy to provide any additional information if needed.

---

### Author Response · Authors · 2024-11-28
**Common Comments**

We sincerely thank you for your time and effort in reviewing our manuscript. Your insightful feedback and constructive suggestions have been invaluable in enhancing the clarity and quality of our work. We are truly grateful for your thoughtful attention to detail.

In response to the reviewers’ questions and to further clarify our methodology, we have revised and updated the manuscript to reflect the final version. Additionally, we have expanded the appendix to include the following new sections:
* Sections D, E, F, G, H, N, O, and P address specific points raised and provide further detailed explanations.
All changes have been highlighted in red text throughout the manuscript to facilitate the review process. With these updates, we have finalized the revised version of our manuscript, which we are now submitting as the final version.

Below, we provide detailed responses to each reviewer’s questions. We sincerely hope these revisions and clarifications fully address your concerns. Should you have any further questions, comments, or suggestions, we would be more than happy to address them.

Once again, we deeply appreciate your valuable time, thoughtful review, and constructive feedback you have provided. Thank you for your expertise and guidance.

---

### Meta-Review · Area_Chair_F5Wy · 2024-12-17

**Metareview:**

The paper investigates the representation of solutions of second-order linear PDEs in Barron space using Green’s functions and provides complexity estimates for their Barron norms. Despite its solid theoretical contributions, the reviewers identified significant weaknesses. These include the restrictive assumptions on PDE coefficients, the limited scope of applicability to simple linear PDEs, and the absence of numerical experiments to support the theoretical results. Concerns were raised about the lack of novelty, as the work primarily extends Chen et al. (2021) and Weinan & Wojtowytsch (2022), and the proofs largely follow established frameworks. The reviewer found that while the work is rigorous, its contributions lack sufficient impact, failing to provide broad insights or practical advancements beyond existing results. The limitations in scope, novelty, and experimental validation ultimately weaken the case for acceptance.

**Additional Comments On Reviewer Discussion:**

During the reviewer discussion, the primary issues focused on the restrictive assumptions, lack of numerical validation, and insufficient extension to more complex PDEs or multi-layer architectures. While the authors provided clarifications and minor revisions, the responses did not fully address the broader concerns regarding applicability and impact. These unresolved points were central to the decision to reject the paper.

---

### Decision · Program_Chairs · 2025-01-22

Reject